## ARTICLES

# Functional ultrasound localization microscopy reveals brain-wide neurovascular activity on a microscopic scale

Noémi Renaudin, Charlie Demené, Alexandre Dizeux, Nathalie Ialy-Radio, Sophie Pezet and Mickael Tanter ✉

The advent of neuroimaging has increased our understanding of brain function. While most brain-wide functional imaging modalities exploit neurovascular coupling to map brain activity at millimeter resolutions, the recording of functional responses at microscopic scale in mammals remains the privilege of invasive electrophysiological or optical approaches, but is mostly restricted to either the cortical surface or the vicinity of implanted sensors. Ultrasound localization microscopy (ULM) has achieved transcranial imaging of cerebrovascular flow, up to micrometre scales, by localizing intravenously injected microbubbles; however, the long acquisition time required to detect microbubbles within microscopic vessels has so far restricted ULM application mainly to microvasculature structural imaging. Here we show how ULM can be modified to quantify functional hyperemia dynamically during brain activation reaching a 6.5-μm spatial and 1-s temporal resolution in deep regions of the rat brain.

Obtaining quantitative information on the function and dysfunction of organs across multiple scales is a challenge of biomedical imaging, as diseases initially emerge locally at the cellular level deep within organs before eliciting large-scale and delayed observable symptoms. In the brain, the interaction between cells and their supplying vessels reaches a high complexity, making such imaging paramount. To support the metabolic demand of billions of neurons, the cerebrovascular system has evolved into a multiscale network, ensuring finely regulated blood supply through precise spatiotemporal modulation of cerebral blood flow (CBF), a phenomenon called neurovascular coupling (NVC)[1–5]. NVC is a fundamental mechanism of brain function and any alteration in this interplay between neurons and vessels is tightly linked to cerebral dysfunction[5]. The majority of neuroimaging modalities[6–12] exploit this NVC by measuring local changes in blood flow or oxygenation during neural activity.

These brain-wide imaging modalities map the activation sites with resolutions ranging from hundreds of micrometres up to the millimeter scale. At such mesoscopic scale, each voxel involves a large number of vessels, ranging from arterioles to capillaries and venules. These compartments exhibit different responses during neurovascular coupling, participating in the complex interpretation of the functional response measured by mesoscopic imaging, such as the blood-oxygen-level-dependent (BOLD) signal for functional magnetic resonance imaging (fMRI)[13] or the Power Doppler signal for functional ultrasound[14]. These different signatures can be interpreted using optical imaging providing information on neurovascular coupling at the microscopic scale, but within a limited field of view[1,15–17]. Such local interpretation fails to account for the large-scale information on the global vascular system architecture to which they belong. The heterogeneities of the neurovascular response in different brain regions[18], or the debated concept of the neurovascular module in the somatosensory cortex[19–21], are some

examples of such large-scale interactions. The contribution of upstream and downstream vascular segments and the major influence of systemic factors on neurovascular function suggested that there is no replicable neurovascular unit, but rather a complex of diverse neurovascular modules. This coordinated interaction of intracerebral microvascular events with larger vessels has recently led to a call for a complete revisiting of the concept of the neurovascular unit introduced in 2001 (ref. [22]). These examples highlight the lack of a brain-wide functional neuroimaging modality reaching the microscopic scale.

Drawing on optical super-resolution techniques[23], fast imaging of intravenously injected microbubbles (MBs) enabled ultrasound imaging to overcome the fundamental trade-off between spatial resolution and penetration depth[24]. ULM overcomes the conventional diffraction limit with two orders of magnitude by localizing and tracking millions of MBs flowing in the blood circulation. It has led to deep microangiographic imaging both in rodents[24–27] and in the clinic[28,29]. Yet, MB displacements are driven by cerebrovascular perfusion, resulting in long acquisition times for obtaining a detailed map of the microvasculature[30]: sensitivity to microscopic vessels (~5–10 μm) comes at the cost of extensive acquisition times (~minutes). This trade-off has limited ULM to morphological imaging of the vascular flow, providing highly resolved but static maps of local hemodynamics or the differentiation of diastolic and systolic hemodynamics[28,31,32].

Here, we demonstrate that functional ULM (fULM) can measure brain-wide vascular activity dynamically at a microscopic scale during task-evoked activity in rodents. We show that fULM provides local estimates of multiple parameters to characterize vascular dynamics such as MB flow, speed and vessel diameters. Additionally, it can decipher the spatial extent and influence of specific vascular compartments of the vascular tree or different vascular arbors during brain activity.

Institute Physics for Medicine Paris, INSERM U1273, ESPCI PSL Paris, CNRS UMR 8631, PSL Research University, Paris, France.
✉e-mail: Mickael.tanter@espci.fr

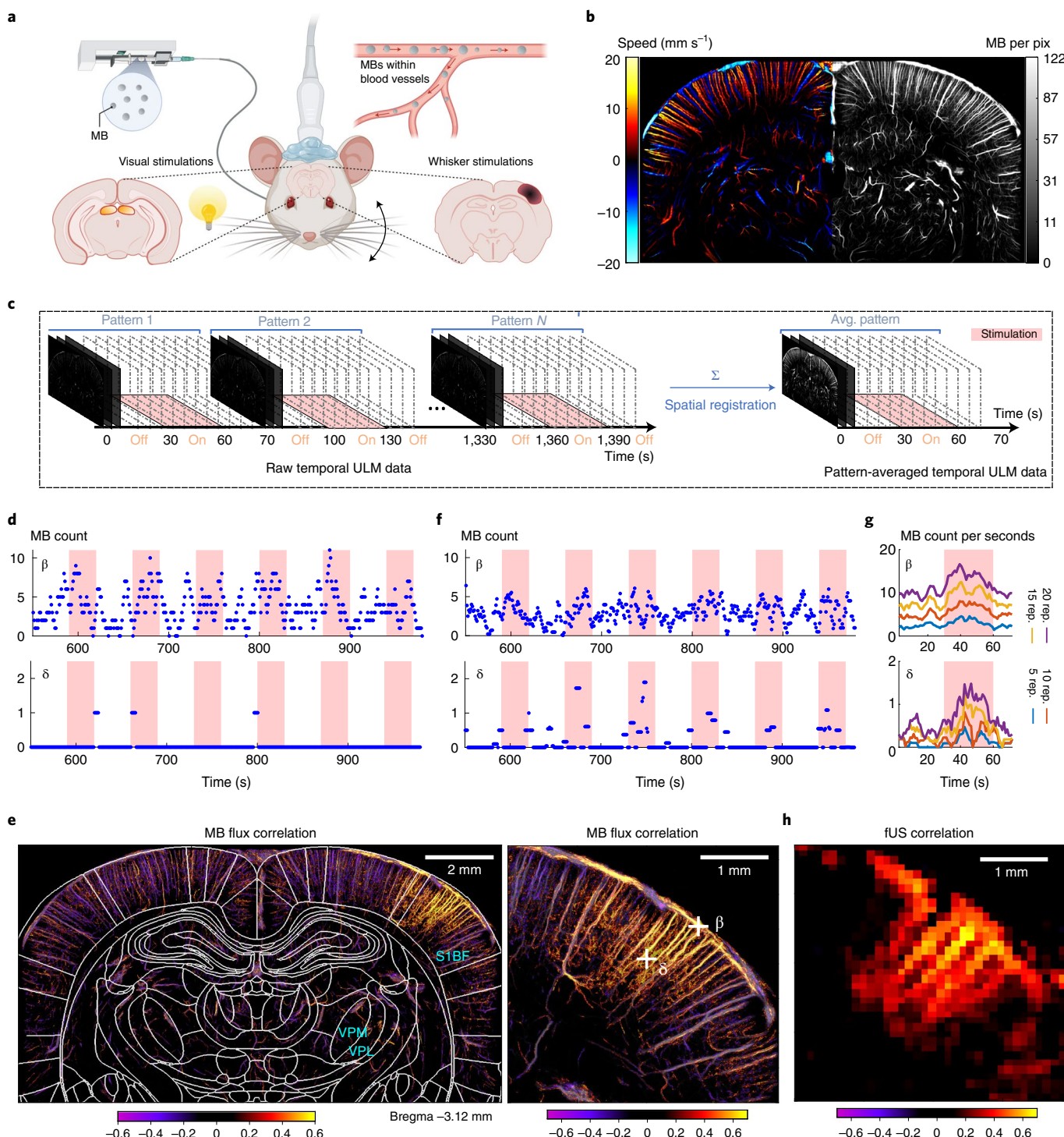

**Fig. 1 | Functional ULM reveals brain-wide hyperemia at a microscopic scale during brain activation. a**, Schematic of the experimental setup for ULM brain imaging through a coronal plane during whisker or visual stimulations in an anesthetized rat receiving a continuous intravenous injection of MBs. **b**, Blood velocity (left) and MB count ULM maps (right) of the rat brain vasculature at 6.5-μm resolution ($n = 7$ experiments). **c**, Temporal rasterization scheme to create dynamic ULM data. **d**, Time courses of MB count for a pixel in a large (pial vessel, β) and smaller blood vessel (first-order branching after descending arteriole, δ), illustrated in **g**. **e**, Pearson correlation coefficient computed between stimulation pattern and MB flux signal, in the whole-brain slice imaged (left) and zoomed in the activated barrel cortex (right). The map is overlaid with rat brain atlas[52] at Bregma $= -3.12$ mm; $n = 4$ experiments. VPL, ventro-postero-lateral thalamic nuclei. **f**, Time courses of MB count for the same pixels after spatial registration. **g**, Time courses of MB flux for the same pixels after pattern summation and division by window length. The time courses are given for an increasing number of pattern repetitions. **h**, Functional correlation map as obtained in **e**, but using conventional fUS imaging (ultrafast Doppler imaging without any MB injection), $n = 4$ experiments.

## fULM exploits data accumulation during repeated task-evoked stimuli

The current study presents both an experimental protocol and a dedicated data processing pipeline for dynamic imaging of functional hyperemia at microscopic scale. We combine sensory stimulations (whiskers deflections or visual stimulations) in anesthetized rats with stable and continuous injection of MBs (Fig. 1a and Extended Data Fig. 1). Instead of using the whole acquisition dataset to compute static ULM two-dimensional (2D) maps of total MB count or average speed (Fig. 1b), we use a temporal sliding window (typically, 5 s with a 1-s step) to construct dynamic maps (Fig. 1c). The temporal window for MB data accumulation is short compared to the traditional accumulation time, leading to sparse temporal signals, especially in first- and second-order capillaries detected by ULM. For example, the temporal signal (Fig. 1d) from pixel β in Fig. 1e within a pial blood vessel exhibits a fluctuating number of MBs between 0 and 10 every 5 s, increasing during whisker stimulations. In a smaller vessel, for a typical pixel such as δ in Fig. 1e located within a first-order branch of a penetrating arteriole, the signal is sparse as the flow is too small to derive a MB detection probability >1 every 5 s; however, the detection of individual MBs occurs mainly during stimulation. At a micrometre scale, slow-motion drift cannot be avoided during the whole acquisition process (>20 min) and is corrected through spatial registration. This motion correction spreads an individual MB detection over several neighboring pixels, leading to a smoother final temporal signal (Fig. 1f).

To obtain non-sparse data and increase signal sensitivity, we took advantage of the stimulus repetition and combine equivalent time points in each pattern to create ULM 2D dynamic maps representative of the stimulus pattern (Fig. 1c). As the number of patterns increases, sensitivity increases and the signal sparsity decreases (Fig. 1g). In pixel δ, using 20 repetitions, the flux reaches 1.4 MB s⁻¹ during stimulation, versus 0.4 MB s⁻¹ on average during baseline activity. As pixel δ describes a vessel with a slower flow compared to pixel β (resulting in a lower MB detection probability), the number of repetitions necessary to obtain non-sparse signals is higher. Temporal profiles for additional pixels are shown in Extended Data Fig. 2.

The Pearson correlation between each pixel pattern-averaged temporal ULM signal and the stimulation paradigm provides the map of functional hyperemia both in cortical and subcortical areas at a 6.5-µm resolution, corresponding to barrel field (S1BF) and ventro-posterio-median (VPM) thalamic nucleus for whisker stimulations (Fig. 1e and Supplementary Video 1) and superior colliculus (SC) for visual stimulations. The spatial resolution is 16-fold better than that achieved with functional ultrasound (fUS) imaging (Fig. 1h).

We quantified the temporal hemodynamic responses during whisker stimulations for $n = 4$ rats and during visual stimulations for $n = 3$ rats (Extended Data Fig. 3). The MB flux reached $+17 \pm 5\%$ in S1BF, $+11 \pm 3\%$ in VPM and $+20 \pm 8\%$ in SC (mean $\pm$ s.e.m.). The velocity reached $+4.7 \pm 2.1\%$ in S1BF $+3.7 \pm 4.9\%$ in VPM, $+4.4 \pm 1.9\%$ in SC (mean $\pm$ s.e.m.). The time profiles of MB flux changes were similar to the corresponding fUS signal. For example, in SC, the MB flux reached a peak, followed by a lower plateau, whereas the MB speed did not exhibit an initial peak.

## fULM unravels the contributions of different vascular compartments

Building on microscopic resolution of fULM, data quantification can be pushed further by distinguishing different vascular compartments (Fig. 2a), a relevant approach as the mechanisms involved in functional hyperemia vary along the vascular tree[33,34]. The dynamic histograms of MB velocities as well as MB flow and velocity time courses (Fig. 2b,c and Extended Data Fig. 4) reveal the highest relative increase in MB flow during activation in intraparenchymal vessels compared to larger vessels such as penetrating arterioles or pial arterioles ($+49 \pm 9\%$ against $+36 \pm 4\%$ for venules, $+32 \pm 7\%$ for penetrating arterioles and $+26 \pm 3\%$ for pial vessels, mean $\pm$ s.e.m.) confirming that these intraparenchymal vessels are the most important contributors to the neurovascular coupling. Depending on the compartment, the responses differed both in shape and amplitude. Results were reproducible on different sets of stimulations (inter-trials s.e.m. in MB flow and speed, respectively, were 6.3% and 0.8% for arterioles, 5.0% and 1.4% for venules, 3.0% and 1.9% for pial vessels and 8.5% and 1.4% for intraparenchymal vessels).

Involvement of each blood vessel during functional hyperemia can also be quantitatively examined, along their entire depth (Fig. 2d–h). In a representative arteriole and venule within the activated barrel, we quantified increases in MB count, speed and diameter ($+38\%$, $+24\%$ and $+37\%$, respectively for the arteriole, $+60\%$, $+30\%$ and $+22\%$ for the venule, at 200 µm below the pia), whereas we did not observe any variation in controls. We also introduced a 'perfusion' and 'drainage area index' to quantify further the involvement of each individual blood vessel (Extended Data Fig. 5 and Supplementary Video 2). They increased by 28% and 54% during stimulation for the representative arteriole and venule, respectively. Due to the large field of view of our imaging modality, these quantitative analyses can be obtained simultaneously for every vessel across the whole rat brain slice image, even in deep cortical structures such as the thalamus for whisker stimulations and SC for visual stimulations (Fig. 3 and Extended Data Fig. 6).

To quantify these effects observed on representative vessels, we analyzed $n = 20$ arterioles and $n = 18$ venules from the activated S1BF of four rats (Extended Data Fig. 7). For barrel arteriolar profiles (Fig. 2i,j) as deep as 400 µm below the pial surface, the quantitative fULM estimates of blood speed ($v_{moy} \sim 14 \pm 1$ mm s⁻¹) and diameter ($D_{moy} \sim 26 \pm 1$ µm) at rest, as well as their relative changes during activation ($\Delta v = +11 \pm 2\%$ and $\Delta D = +19 \pm 3\%$), are consistent with measurements made using two-photon microscopy in the

**Fig. 2 | Super-resolved quantification applied to MB trajectories reveals increased MB flux, speed and vessel diameter in arterioles and venules of the activated barrel cortex during functional hyperemia. a**, Subdivision of the barrel cortex into penetrating arterioles, venules, pial vessels and intraparenchymal vessels based on the super-resolved ULM maps. **b**, Dynamic histograms of the MB velocity distribution in the compartments defined in **a** during whisker stimulations (stim, $n = 40$ stimuli). **c**, Mean MB flow and speed ($\pm$s.e.m.) from $n = 4$ time courses obtained on ten stimulations each, either expressed as absolute value for each compartment (left), or as relative to baseline (right). **d–g**, Longitudinal profiles of MB count, speed and diameter during rest and stimulation periods, along an arteriole (**d,e**) or venule (**f,g**), activated site (**e,g**) or control site (**d,f**). Labeling of all MB trajectories passing through the chosen white segment at the entry of the penetrating arteriole or venule. Quantification of this perfusion of drainage area was performed using this selective set of MBs. For each blood vessel, those area for rest and stimulation periods are displayed on ULM MB count maps. **h**, Changes during whisker stimulation relative to rest in the MB count and speed for the blood vessels analyzed in **d–g**. **i,j**, Analysis on $n = 20$ arterioles (20 in the activated barrels and 20 controls) and $n = 18$ venules (18 in the activated barrels and 18 controls) transversal sections from $n = 4$ rats at a depth <400 µm from pial vessels. MB count (**i**) and speed (**j**) transversal profiles shown as mean $\pm$ s.e.m. Percentage variation relative to rest and the results from a two-sided Wilcoxon signed-rank test on this variation (null hypothesis, distribution with median equal to zero; NS, not significant, *$P \leq 0.05$, **$P \leq 0.01$, ***$P \leq 0.001$) are given for the max MB count (activated arterioles, $P = 9 \times 10^{-5}$; control arterioles, $P = 0.91$; activated venules, $P = 9 \times 10^{-4}$; control arterioles, $P = 0.98$) and speed ($P = 9 \times 10^{-5}$; $P = 0.55$; $P = 3 \times 10^{-4}$; $P = 0.012$); $n = 4$ experiments (**a–j**).

cortex (respectively $v_{moy} \sim 9\,mm\,s^{-1}$, $D_{moy} \sim 18\,\mu m$, $\Delta v = +17\%$ and $\Delta D = +12\%$)[33] and the olfactory bulb (respectively $v_{moy} \sim 13\,mm\,s^{-1}$, $D_{moy} \sim 20\,\mu m$, $\Delta v = +15\%$ and $\Delta D = +20\%$)[34]. The perfusion area increased by $21 \pm 4\%$. Deeper in the cortex (>600 μm), this increase in MB count was still $+27 \pm 6\%$, but only $+4 \pm 1\%$ for the velocity, whereas we observed a $+22 \pm 4\%$ increase of the diameter.

The speed increase between the two depths was significantly different ($P = 0.002$, paired two-sided Wilcoxon signed-rank test), as exemplified with the arteriole in Fig. 2d showing an increase in speed limited to the first hundreds of micrometres. Mapping the dilatation and constriction of vessels is also possible (Extended Data Fig. 7c,d).

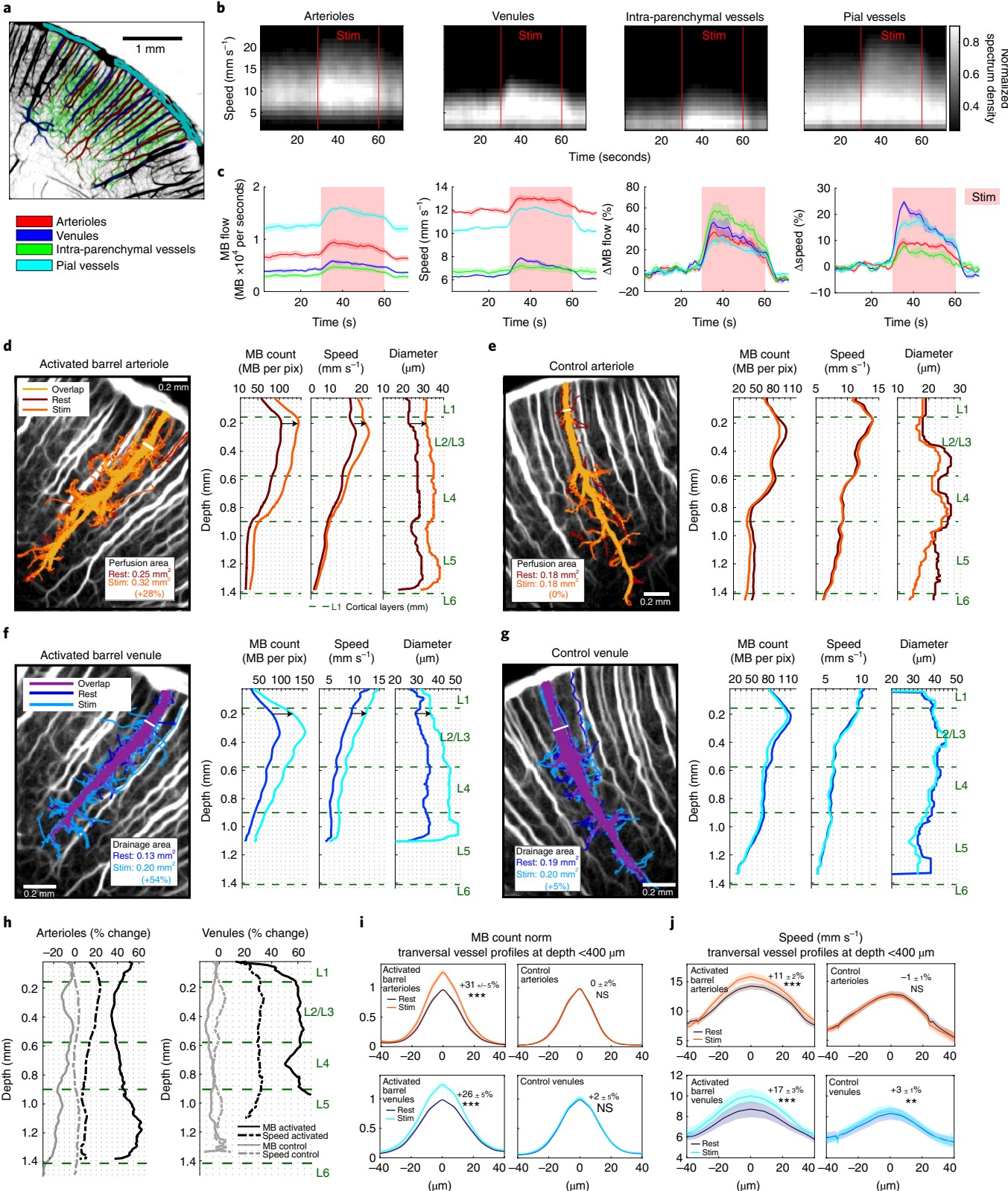

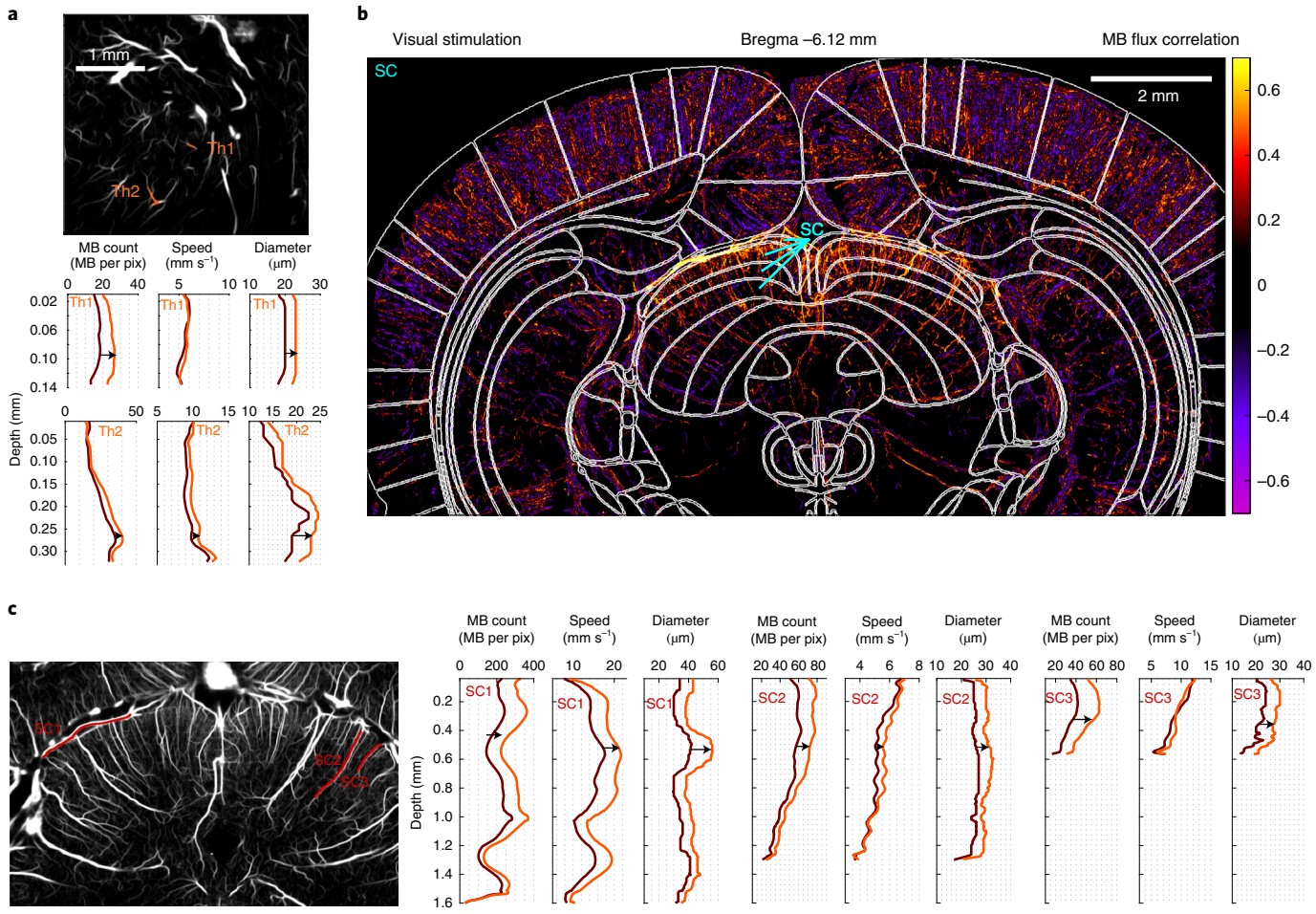

**Fig. 3 | Functional ULM reveals activation in subcortical structures such as VPM and VPL after whisker stimulation and SC after visual stimulation.** **a**, Selection of two blood vessels (orange, Th1 and Th2) within the activated thalamus displayed on an ULM MB count map. MB count, speed and diameter quantifications for rest and stimulation periods for the two blood vessels; n = 4 experiments. **b**, MB flux correlation map after visual stimulations overlaid with rat brain atlas[52] at Bregma = −6.12 mm. **c**, Selection of three blood vessels (red, SC1, SC2, and SC3) within the activated SC displayed on an ULM MB count map. MB count, speed and diameter quantifications for rest and stimulation periods for the three blood vessels; n = 3 experiments (**b**,**c**).

## Singular value decomposition further expands fULM capabilities

To provide quantitative maps of brain activation, we found that singular value decomposition (SVD) of the pattern-averaged three-dimensional (3D) ULM matrix can isolate functional hyperemia in one singular mode, whereas singular vectors of higher singular values quantify local variations in MB concentration. For whisker stimulation, the first spatial singular mode depicts the vasculature at baseline level (Extended Data Fig. 8a), whereas the spatial activation pattern is isolated in the second spatial singular mode (Fig. 4a). The automatic detection of the singular mode corresponding to the stimulus is obtained in all configurations using the scalar product between each temporal singular vector and the stimulation pattern (Extended Data Fig. 8b–h). The associated temporal singular vectors are displayed in Fig. 4b. We applied SVD both to the MB flux and the velocity matrices, which yielded quantitative imaging of the increase in MB count and speed (Fig. 4c,d). We also mapped maps via the SVD the relative increase in MB count (Fig. 4e), which depicted a stronger relative increase within intraparenchymal vessels compared to penetrating branches, confirming earlier results (Fig. 2c).

The ability of SVD to spatiotemporally detect small stimulus-related signals is dependent on the number of pattern repetitions.

Below five stimuli, the SVD processing cannot detect and locate the activation, but it converges toward a stable solution after ten repetitions (Extended Data Fig. 9). For sufficiently strong activations, SVD can even isolate activity after individual stimulation when applied to the raw sparse ULM data without any pattern summation (Fig. 4f,g). As the diversity of MB temporal behaviors is low compared to the global number of spatiotemporal MB detections (Supplementary Fig. 1), the SVD activation map (fourth singular mode; Extended Data Fig. 8h) provides local information even in pixels containing a small number of detections (Fig. 4f). This map is consistent with pattern-averaged results (Fig. 4c), but it extracts additional information as the time course of every single trial is retrieved (Fig. 4g). The first three singular vectors describe the basal level of MBs and the small decay in MB concentration after the injection at the beginning of the experiment (Extended Data Fig. 1).

Finally, this SVD analysis stands up to more challenging experimental conditions. For bolus injections (Fig. 4h−m), it discriminates bolus against stimulus-related variations, isolating the baseline, injection pattern and activation signal, respectively in the first, second and third modes (Fig. 4m). The results are similar using a continuous injection versus a bolus injection, showing the robustness of the SVD analysis. Transcranial fULM is also conceivable (Supplementary Fig. 2).

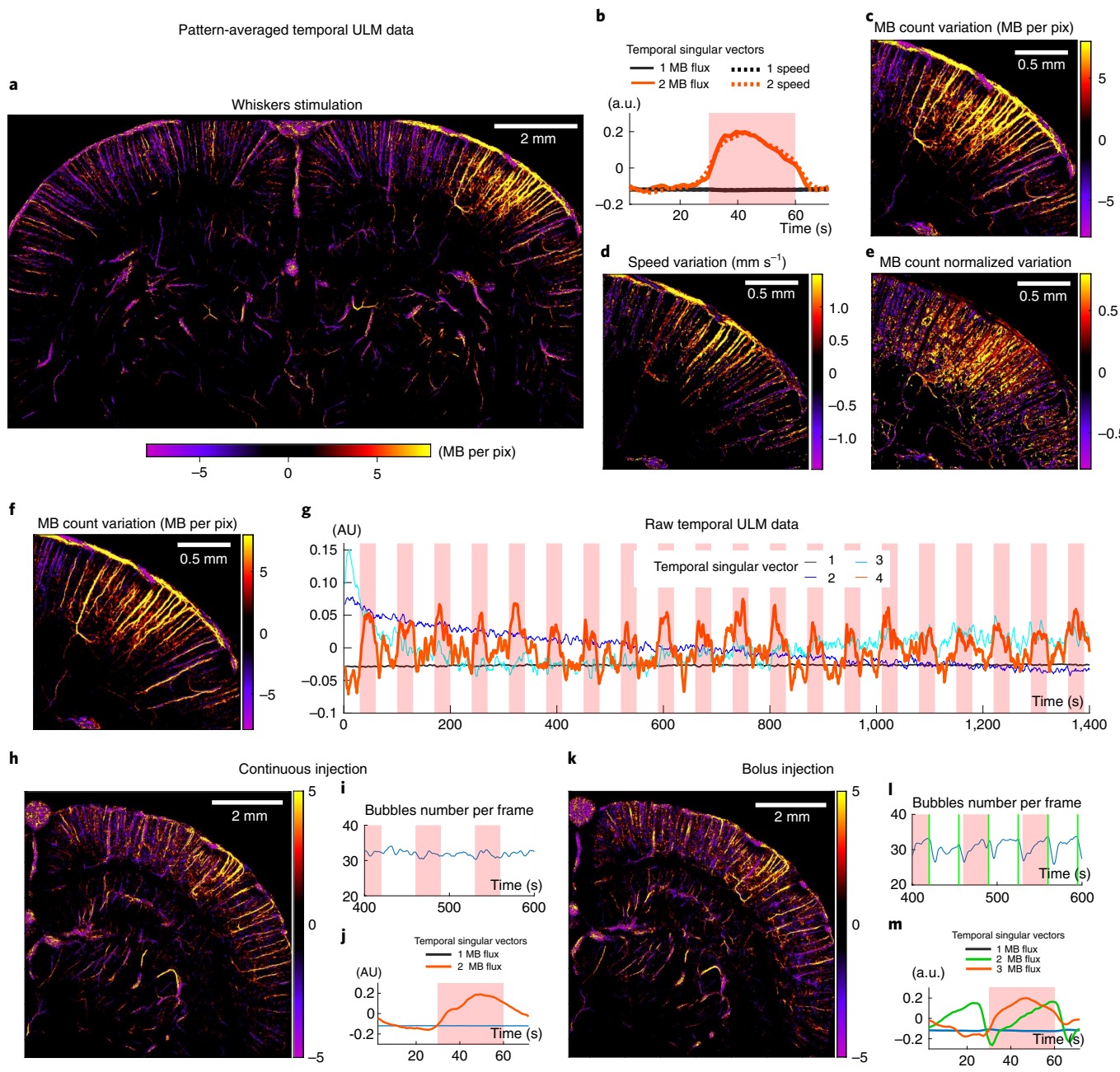

**Fig. 4 | Singular value decomposition of sparse dynamic ULM data extracts the spatial and temporal profile of multiple parameters during activation.**
**a,b**, SVD analysis applied to pattern-averaged data: MB count variation map during stimulation (**a**) and associated temporal singular vectors (**b**).
**c–e**, Zooming in on the somatosensory barrel cortex, absolute MB count variation (**c**), speed variation (**d**) and relative MB count variation (**e**). **f,g**, SVD analysis applied to the raw temporal ULM matrix (without any pattern summation, sparse signal): stimulation spatial singular vector (**f**) and corresponding temporal singular vectors (**g**). **h–m**, SVD analysis in continuous versus bolus injections experiments. These experiments were performed in the same rat, using either a continuous injection (**i**) or bolus injections every 35 s (**l**). Results show the spatial singular vector corresponding to stimulation (**h,k**) and the first SVD temporal singular vectors (**j,m**). $n = 4$ experiments (**a–e**), single micrograph (**f–m**).

## Spatial resolution, temporal resolution and sensitivity to slow flow

As is the case for other imaging modalities, fULM makes a compromise between spatial resolution, temporal resolution and signal-to-noise ratio (SNR) and these parameters cannot be defined independently.

The dynamic ULM maps were here reconstructed on a $6.25 \times 6.875\,\mu m$ grid. We confirmed this spatial resolution by quantifying the statistic distribution of velocities in neighboring pixels (Extended Data Fig. 10a–d) corresponding to one time point of the pattern-averaged data[24,28]. Additionally, we estimated tissue pulsatility (Extended Data Fig. 10e–i) and found it to be $2.2 \pm 1.4\,\mu m$ (mean $\pm$ s.d.) and $1.6 \pm 1.3\,\mu m$ for respiratory and cardiac motion, respectively. This vascular pulsatility can be corrected as shown previously[35]. It should also be noted that these motion artifacts are partially canceled during the initial SVD clutter filtering of raw data (Extended Data Fig. 10j). Finally, to correct for unavoidable motion drift occurring on time scales much slower (10 s) than the cardiac or breathing time scales, we also added intensity-based spatial registration (translation transformation) based on ULM MB count maps.

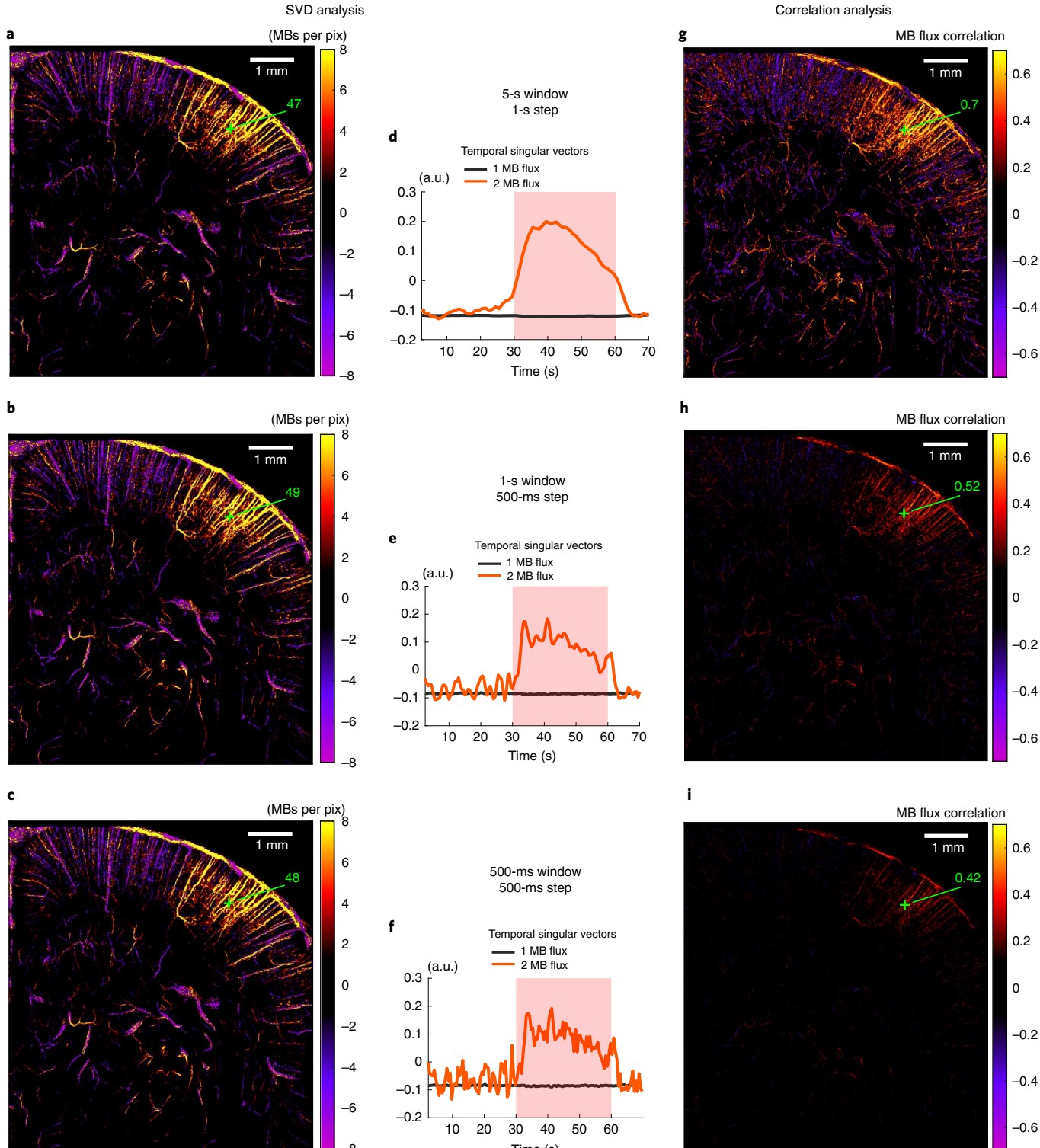

**Fig. 5 | Temporal resolution. a–c**, SVD applied to MB flux. Spatial singular vector corresponding to functional hyperemia (second singular mode for **a–c**) for different temporal resolutions: 5-s window with 1-s step (**a**); 1-s window with a 0.5-s step (**b**); and 0.5-s with a 0.5-s step (**c**). **d–f**, Corresponding temporal singular vector. **g–i**, Results for the same experiment and the same temporal resolutions as in **a–c** but using correlation analysis (Pearson correlation coefficient computed between stimulation pattern and MB flux signal); $n = 7$ experiments (**a–i**).

So far, we have presented data using a 5-s window with a 1-s step and 20 stimulations, leading to each ULM image being created with a 100-s accumulation time; however, the sliding window can be tailored to refine the temporal analysis. The activation map with a 1-s window and 0.5-s step does not degrade using the SVD analysis, showing that the SVD analysis is more robust to injection noise than the correlation analysis (Fig. 5). We also show results for a 500-ms window, where there is a risk of losing sensitivity to the smallest blood vessels by diminishing the temporal step too much. Furthermore, we performed additional analyses where we tailored the window length.

For example, we used a 2-s window with a 500-ms step to study a shorter stimulation (5-s visual stimulus; Supplementary Fig. 3) or we repeated an earlier analysis (Fig. 2c) using a 1-s sliding window with a 500-ms step (Supplementary Fig. 4).

Sensitivity to slow blood flow is high even at depth as microbubbles flowing at least down to 1 mm s$^{-1}$ are detectable (Supplementary Fig. 5). Such low speeds correspond to vessel diameters going down to 10 µm[36]. The SNR during brain activation is more complex to define as it depends on the location of the voxel. If the voxel of interest is within an arteriole with typical 30–60-µm diameter, several MBs can be detected every second leading to a high SNR. Such SNR is sufficient to perform single trial measurements (Fig. 1e voxel β). If the voxel in question is within a capillary, accumulation over several stimuli is required but finally leads to a MB count increase per second relative to baseline exceeding several times the baseline s.d. (Fig. 1f; MB s$^{-1}$ increase / baseline s.d. = 5.2 and 4.1 in voxels β and δ, respectively).

## Discussion

fULM offers a quantitative look at the cerebral microcirculatory network and its hemodynamic changes by combining a brain-wide spatial extent with a microscopic resolution and a temporal resolution (1 s) compatible with neurofunctional imaging.

To address the trade-off between sensitivity and minute acquisition times, we increased the number of MBs detected per second based on the repetition of stimuli during ULM. To better extract spatiotemporal dynamics due to brain activation, we applied SVD to the spatiotemporal ULM data. This approach provides both the spatial map and temporal profile of activation in a unique singular mode. It isolates slow trends of MB concentration during continuous injections and large variations in the case of successive bolus injections. Despite the sparsity of data in small vessels, it can also be efficient for non-averaged data and provides spatiotemporal activation patterns on a single-trial basis.

The achieved spatiotemporal resolution enables fULM to image different vascular compartments simultaneously in the whole brain and to discriminate their respective contributions, in particular in the precapillary arterioles known to have a major contribution to vascular changes during neuronal activity. Furthermore, tracking the long-range trajectory of MBs in a Lagrangian representation of the MB flow provides a way to disentangle the spatial extent of downstream or upstream vessels depending on an individual arteriolar input or venule output. Finally, as each individual MB serves as a hemodynamic sensor, the estimation of mean quantitative parameters, such as MB flow speed or flux, relies on large numbers of samples, leading to a high accuracy and small s.e.m.

Recent works have demonstrated that CBF is mainly controlled by arterial smooth muscle cells in arterioles greater than 10 µm (detectable by fULM), but not by capillary pericytes[37,38]. The arterioles and first-order capillaries dilate first and form the primary functional unit[20]. These precapillary arterioles can be detected by fULM. They involve smooth muscle cells and mesh pericytes known to play a major role in the neurovascular coupling[34,39–41]. Precapillary sphincters at the transition between the penetrating arteriole and first-order capillary[33] generate the largest changes in the cerebrovascular flow resistance, thereby controlling capillary flow while protecting brain tissue from adverse pressure fluctuations. fULM brings a complementary insight to these previous results by showing that the relative increase in MB flow is greater in intra-parenchymal vessels (first-order capillary and lower) rather than in arterioles. fULM also confirms depth-dependent characteristics for blood flow and speed in penetrating arterioles at baseline[21,42]. During activation, fULM further highlights a depth-dependent variation in blood speed. It also quantifies large increases of MB flux, blood speed and diameter in venules during activation, consistent with fMRI and two-photon studies reporting considerable increases in venous blood flow during long stimulations[43,44].

fULM also has limitations. Wide-field optical imaging can reach similar and even higher spatiotemporal resolution at the cortical surface[45]. Compared to optics or photoacoustics, fULM has only access to vascular and hemodynamic information. Although we could identify first- and second-order capillaries using fULM, the detection limit within the vascular tree is not clear and we cannot rely on cell-type markers but can only base our segmentations on morphometric descriptors. This latter description has, however, been used previously[41,46]. We have applied fULM in anesthetized rats but the development of fULM in awake animals using head-fixed configurations may not be out of reach[47,48]. To this end, we have conducted preliminary experiments of transcranial ULM in mice using MB injections in the tail at a slower 1 ml h$^{-1}$ rate and obtained high-quality images.

The skull bone also affects the image quality in ULM. In mice, transcranial ULM can be performed[48]. In rats, transcranial fULM would benefit from aberration correction techniques to avoid shadowed regions, similarly to what was performed recently in humans[28]. Recent works already show convincing transcranial ULM in rats[49].

The influence of the injected volume on brain physiological parameters should also be carefully investigated. Our small injection volume (1.1 ml during 20 min) and slow 3.5 ml h$^{-1}$ rate is six times below the maximal dose of the international recommendations and guidance for injections[50]. It could be further decreased as we applied restrictive conditions for MB detection and tracking, leading to a small number of preserved MB (typically less than 40% of globally detected MBs) per ultrafast frame.

The need for repetitive stimuli is also a limitation, although mainly for the smallest vessels. In typical 30-µm diameter arterioles, MBs per second are sufficient to provide dynamic fULM without repetition (in single trials). In 10-µm diameter vessels, the required repetitions number is typically $n = 10$, which remains reasonable for whole-brain imaging.

Beyond task-evoked stimuli, the applicability of fULM during spontaneous activity could also provide information on functional brain connectivity. The non-repetitive nature of spontaneous activity could make its mapping with fULM more difficult; however, results obtained by the SVD analysis without any pattern summation inspire hope for success. Finally, the feasibility of 3D brain ultrasound imaging[49,51] also suggests the extension of fULM to 3D imaging.

fULM provides a way to track dynamic changes during brain activation and will offer insights into neural brain circuits, as it provides a tool for the study of functional connectivity, layer-specific cortical activity or neurovascular coupling alterations on a brain-wide scale. Application in mice should be straightforward and paves the way to using genetically modified models and molecular tools relevant for the investigation of neurovascular coupling. fULM could also be implemented in combination with spatial RNA-sequencing techniques to get microvascular assessment with cell-type specificity. Application of fULM in humans requires further refinements, but the recent demonstration of transcranial ULM[28] in humans raises hope for its future clinical implementation.

## Online content

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

## Methods

**Animals.** *Animals.* All experiments were performed in compliance with the European Community Council Directive of September 22, 2010 (010/63/UE) and the local ethics committee (Comité d'éthique en matière d'expérimentation animale no. 59, 'Paris Centre et Sud', project no. 2017–23) and with ARRIVE guidelines. Accordingly, the number of animals in our study was kept to the necessary minimum following the 3Rs (reduce, refine and replace) guidelines. Experiments were performed on $n = 10$ male (aged 7–9 weeks) Sprague–Dawley rats (Janvier Labs), weighing 200–300 g at the beginning of the experiments. Animals (housed two per cage) arrived in the laboratory 1 week before the beginning of the experiment and were maintained under controlled conditions ($22 \pm 1\,°C$, $60 \pm 10\%$ relative humidity, 12/12-h light/dark cycle, food and water ad libitum). All animals included in this study were untreated and were used randomly in the various experiments.

*Surgical procedure and preparation for imaging.* Under deep anesthesia (intraperitoneal (i.p.) bolus of medetomidine (Domitor, $0.4\,\mathrm{mg\,kg^{-1}}$) and ketamine (Imalgène, $40\,\mathrm{mg\,kg^{-1}}$)), a catheter filled with saline was inserted in the jugular vein of the rat before positioning the animal on the stereotaxic frame (Fig. 1a; created with BioRender.com). A craniotomy (removal of the skull) was then performed between Bregma and Lambda. This window enables a large part of the brain to be scanned, the part under which the S1BF and SC are located. During the surgical procedure and the imaging session, the animals' body temperature was kept at $37\,°C$ using a heating blanket and an intrarectal probe (Physitemp). Around 45 min after induction (when the craniotomy was completed), the anesthesia was maintained but reduced, using subcutaneous perfusion of medetomidine ($0.1\,\mathrm{mg\,kg^{-1}\,h^{-1}}$) and ketamine ($12.5\,\mathrm{mg\,kg^{-1}\,h^{-1}}$) with a syringe pump. The heart and respiratory frequencies were monitored continuously to ensure stability of the anesthesia (Labchart, AD Instruments). Each imaging session lasted between 2 and 4 h. Two milliliters of saline were gently dropped on to the brain (the dura mater was kept intact), followed by echographic gel (Dexco Medical). The ultrasonic probe was then positioned just above the window, using a three-axis motorized system on which the ultrasound probe was fixed.

For the transcranial experiment, no craniotomy was performed but the skin above the skull was removed before dropping the saline and the echographic gel. The images in transcranial experiments present some shadowed areas caused by a well-known stripe artifact[53]: the curvature in the internal surface of the skull where major pial vessels run introduce aberration of the ultrasound waves, producing shadowed areas underneath (Supplementary Fig. 2).

*Stimulation protocol.* For the whisker stimulus, facial whiskers were stimulated using a custom-made mechanical stimulator, brushing the whiskers at a 10-Hz frequency. For the visual stimulus, a white LED, positioned at 10 cm from the head, delivered flashes at a 3-Hz repetition frequency.

Both types of stimulations were triggered during the imaging sequence by a microcontroller (Arduino Uno) controlling a servomotor (or an LED), ensuring synchronization between stimulation and fUS or ULM recording, as well as precision and reproducibility during stimulation. The stimulation pattern (unless stated otherwise) for visual and whisker stimulations was 30 s rest, 30 s stimulation and 10 s rest. The short visual stimulation was 15 s rest, 5 s stimulation and 5 s rest. The number of stimulations was set at 5 for fUS imaging experiments and to 20 to 60 for ULM experiments (20 if not stated otherwise).

**Functional ultrasound imaging.** *fUS imaging acquisition and processing.* fUS imaging was performed using rapid acquisition of ultrasensitive 2D Power Doppler images of the rat brain using a linear ultrasound probe (128 elements, 15.625 MHz, 110 μm pitch, 8 mm elevation focus, Vermon) connected to an ultrafast ultrasound scanner (Iconeus, 128 channels, 62.5 MHz sampling rate) driven with Neuroscan live acquisition software (v.1.3, Iconeus and INSERM Accelerator of Technological Research in Biomedical Ultrasound). The pulse shape corresponded to two periods of sinusoids, the transmit voltage was 25 V and the mechanical index was 0.44. For each Power Doppler image, 200 frames were acquired at a 500-Hz frame rate, each frame being a compound frame acquired via 11 tilted plane wave emissions ($-10°$ to $10°$ with $2°$ steps) fired at a 5,500 Hz pulse repetition frequency. These Power Doppler images were acquired continuously at a 0.4 s frame rate. Image reconstruction was performed using GPU-parallelized delay-and-sum beamforming. Those 200 frames at 500 Hz were then filtered to discard global tissue motion from the signal, using a dedicated spatiotemporal clutter filter[54] based on a SVD of the spatiotemporal raw data; the 60 first singular values were discarded. Blood signal energy (Power Doppler) was computed for each voxel ($100 \times 110 \times \sim400\,\mu m$, the third dimension, called elevation, being slightly dependent of depth) by taking the integral over the 200 time points[11,55]. This Power Doppler signal is known to be proportional to blood volume[56] for a constant hematocrit.

*Functional ultrasound activation maps and region of interest selection.* Correlation maps (Fig. 1h) were calculated as the Pearson's product-moment correlation coefficient $c$ between the stimulation pattern A(t) (slot with 0 for rest periods and 1 for stimulation periods) and Power Doppler PD(t) signal for each pixel:

$$c(x,z) = \frac{\sum_{k=1}^{Nst}\left(\mathrm{PD}\left(x,\,z,\,t_k\right) - \overline{\mathrm{PD}\left(x,z\right)}\right) \times \left(A(t_k) - \bar{A}\right)}{\sqrt{\sum_{k=1}^{Nst}\left(\mathrm{PD}\left(x,\,z,\,t_k\right) - \overline{\mathrm{PD}\left(x,z\right)}\right)^2} \times \sqrt{\sum_{k=1}^{Nst}\left(A\left(t_k\right) - \bar{A}\right)^2}}$$

Power Doppler data were analyzed using a generalized linear model approach, implemented in MATLAB to obtain $z$ scores (Extended Data Figs. 3a,b and 7a) and $P$ value maps. The activated area (corresponding to the primary S1BF, VPM and VPL, and the SC) were chosen as significant pixels in the image with a $P$ value $<0.05$ after Bonferroni correction for multiple comparisons. The control area chosen for whisker stimulation experiments was drawn by symmetry in the contralateral S1BF. For the visual stimulations, the control area was drawn in the area corresponding to the caudal part of the retrosplenial cortex located in the imaging plane (Extended Data Fig. 3a,b).

To obtain temporal responses in specific regions of interest (ROIs), the Power Doppler signal was averaged along the two spatial dimensions of the ROI. The signal was then normalized toward the baseline (Extended Data Fig. 3e,h).

**Functional ultrasound localization microscopy.** For each animal, the fUS imaging experiment (without contrast agents) was first performed. The ULM experiment was then performed in exactly the same imaging plane.

*ULM acquisition and processing.* ULM acquisitions were performed similarly to the methods described previously[28], but using a continuous injection at a $3.5\,\mathrm{ml\,h^{-1}}$ dose of Sonovue (Braco, reconstructed in 5 ml saline, as recommended by the manufacturer), using a push syringe (KD Scientific). The international recommendations for the maximum injection volume for intravenous (i.v.) injections in rats[57] is $20\,\mathrm{ml\,kg^{-1}}$ with a slow injection (at least 3–10-min long). For a 300 g rat, the maximum injection volume should not exceed 6 ml. In our experiment, we had a continuous slow injection ($3.5\,\mathrm{ml\,h^{-1}}$ during 20 min) corresponding to a 1.1 ml injection volume (25% of the maximum dose) at a rate three to eight times slower than the slow injection described in these international recommendations. A magnet was inserted in the syringe to mix the MB solution during the acquisition. Continuous injections enabled a stable number of MBs to be secured for more than 20 min with approximately 30 MBs per ultrasound frame after detection and tracking (Extended Data Fig. 1). The MB s$^{-1}$ baseline does not vary substantially during the 20 min of continuous injections in the different regions of the brain, revealing that the CBF remains stable during the acquisition despite the volume injected. The 1.1 ml injection volume for a 21 ml total blood volume (300 g rat) corresponds to a maximal 5% change in total blood volume due to the injection (which remains limited[58,59]). We also assume the hematocrit to be stable as a change in hematocrit would lead to a change in CBF.

Blocks of 400 compounded frames at a 1,000-Hz frame rate (each frame is a compound image acquired with angles at $-5°$, $-2°$, $0°$, $+2°$ and $+5°$ fired at a 5,000 Hz PRF) were acquired using the same probe (128 elements, 15 MHz, 110 μm pitch, 8 mm elevation focus, Vermon). The pulse shape corresponds to two periods of sinusoids, the transmit voltage is 5 V and the mechanical index is 0.09. Blocks were continuously acquired (no temporal gap between two successive blocks). Beam-formed data were filtered using the SVD spatiotemporal clutter filter[54] to discriminate the ultrasonic signature of individual MBs from tissue signals; the ten first singular values were discarded. Images were interpolated (Lanczos interpolation kernel) down to (probe spatial pitch $/6 \times \lambda/6$). A movie of the resulting images is provided in Supplementary Video 3. A binary mask was built based on the vesselness[60] filtering of this stack of images (3D implementation available on Mathworks file exchange)[61]. MBs were detected as the brightest local maxima with high correlation ($>0.7$) with a typical point-spread function (imaging response of an isolated MB, modeled as a Gaussian spot of axial and lateral dimension of lambda). Sub-pixel maxima localization was then performed using a fast local ($5 \times 5$-pixel neighborhood) second-order polynomial fit. The resulting coordinates were rounded to the chosen pixel size (here $6.875 \times 6.25$ micrometres = initial pixel size $/16$). Tracking of the maxima positions was performed using a classical particle tracking algorithm (simpletracker.m available on Mathworks Tinevez[62], wrapping the MATLAB munkres algorithm implementation of (ref. [63])), with no gap filling and maximal linking distance corresponding to a 100 mm s$^{-1}$ maximum speed. Only tracks with MBs detected in at least ten successive ultrafast frames were selected. The successive positions gathered in one track were used to compute the interframe bubble velocity vector components (along probe $x$ axis and depth $z$ axis) and absolute velocity magnitude. We added a linear spatial interpolation on each track to count one MB detection in every pixel on the MB path. Maps of MB count were computed by counting all the MBs detected in one pixel during the acquisition time; velocity maps were computed as their mean velocity (Fig. 1b). In plane pixel size for image reconstruction is $\sim6.5\,\mu m$ ($6.875 \times 6.25\,\mu m$).

*ULM images construction and data analysis.* During the ULM processing every track was saved with each MB position and its respective time position. Constructing ULM images is performed by selecting a pixel size and by sorting each MB within each pixel. Only pixels accumulating at least five different MB detections during the total acquisition time were considered during the various analyses.

**Quantification and correction of local tissue pulsatility.** We estimated the impact of tissue motion due to cardiac and breathing pulsatility. We performed speckle tracking correlation on 1.4 s blocks of raw IQ data for the animal presented in Fig. 1. Spatially averaged tissue displacements exhibit both cardiac and breathing pulsatility with respective ~180 ms and ~620 ms periods (corresponding to ~330 b.p.m. and ~97 b.p.m. coherent with the literature) (Extended Data Fig. 10a). The mean tissue motion error was estimated for each pixel and is represented in Extended Data Fig. 10b–e. An averaged $2.2 \pm 1.4\,\mu m$ (mean ± s.d.) motion error was found over the imaged area for respiratory motion and $1.6 \pm 1.3\,\mu m$ (mean ± s.d.) for cardiac motion. Such potential tissue motion artifacts would typically be of the same order as the pixel size in our experiments. Nevertheless, it should also be noted than these motion artifacts are in fact canceled during the initial SVD of raw IQ data. To illustrate this, the first temporal singular vectors of the same 2-s block of raw data are presented in Extended Data Fig. 10j. One can notice that the temporal singular vector no. 1 contains the tissue pulsatility in good agreement with the results obtained using speckle tracking correlation in Extended Data Fig. 10e. As these singular vectors are canceled during the SVD filtering process dedicated to cancel tissue signals (the cutoff threshold was set to $n = 10$), the filtered data containing the MB signature enables the at least partial cancelation of this motion artifact signature. Another possible correction strategy could be to use the tissue motion estimates from speckle tracking correlation of raw date to correct the position of MBs before the SVD processing as shown previously[35].

**Correction of slow-motion drift during large acquisition times.** To account for unavoidable motion drift occurring on time scales much slower than the cardiac or breathing time scales, we also added an intensity-based spatial registration (translation transformation) based on 10 s ULM MB count maps (dimension $x$, $z$ and $t$) to correct for any drift during the global acquisition time (>20 min). We used the MATLAB functions imregconfig (monomodal option) and imregtfrom (translation option).

**ULM temporal analysis.** For temporal ULM data construction, a first set of 3D ULM temporal matrices (MB count matrix MB($x,z,t$) and a velocity matrix V($x,z,t$)) was constructed using a sliding window for data accumulation. These take the form M($N_x$, $N_z$, $N_t$) where M($t$) corresponds to a 2D ULM image calculated on ULM data for MB detection time $\in [t - W_t/2 : t + W_t/2]$ with $W_t$ corresponding to the sliding window duration.

This produced a matrix with a third dimension equal to:
$N_t = \frac{Acq_t}{Step_t} = \frac{Nb_{pattern} \times Pattern_t}{Step_t}$

With:

Acq$_t$, whole acquisition duration; step$_t$, step of the sliding window; pattern$_t$, stimulus pattern duration

The pixel value for the MB count matrix is equal to the number of MBs whose trajectory passed through this pixel during the corresponding temporal window. The pixel value for the velocity matrix is given by their mean velocity. This set of temporal ULM matrices is referred to as 'raw temporal ULM data'.

A sliding window of 5-s length with a 1-s sliding step was used unless stated otherwise.

A second set of ULM temporal maps M$^s$($N_x$, $N_z$, $N^s_t$) was then created by accumulating data from equivalent time points with regards to the stimulation pattern. This produced matrices with a third dimension equal to: $N^s_t = \frac{Pattern_t}{Step_t}$

The MB count value was further divided by the window length to get a MB flux value (MB s$^{-1}$).

$MB_F(x, z, t)$

$= \frac{1}{W_t} \sum_{i=0}^{Nb_{Pattern}-1} MB(x, z, t + i \times Pattern_t), \quad t < pattern\_length$

V$^s$($x,z,t$) is the mean velocity of all MB detected at pixel ($x,z$) during $[t + (i-1) \times pattern_t - W_t/2 : t + (i-1) \times pattern_t + W_t/2]$,

This second set of ULM matrices is referred as 'pattern-averaged temporal ULM data'.

To get temporal ULM responses in specific ROIs (Fig. 2c, Extended Data Figs. 2e,f, 3c,d,f,g and 4c and Supplementary Fig. 4), M or M$^s$ matrices were summed along the two spatial dimensions of the ROI. The same approach was performed for velocity (Vs) signals, but using averaging. The signal was then normalized by the baseline to get relative variations.

For quantification of the spatial resolution, similar to previous work[24,28], the spatial resolution was evaluated by quantifying the statistic distribution of velocities for the MB population detected in a single pixel of $6.875 \times 6.25\,\mu m$ in one map of the ULM temporal maps M$^s$($N_x$, $N_z$, $N^s_t$) and comparing it with the same distribution in the neighboring pixel. For one cortical and one deeper vessel (Extended Data Fig. 10a–d), MB velocities flowing through a transversal section of the vessels during time windows were gathered in bins corresponding to each $6.875 \times 6.25\,\mu m$ pixel. A two-sampled Student's $t$-test was applied between each consecutive bin.

For activation maps using correlation analysis, maps (Figs. 1g, 3b and 5g–i and Extended Data Fig. 2d) were calculated as the Pearson's product-moment

correlation coefficient $c$ between the stimulation pattern A($t$) (slot with 0 for rest periods and 1 for stimulation periods) and $M_s(t)$ for each pixel:

$$c(x,z) = \frac{\sum_{k=1}^{Nst} \left( M_s(x, z, t_k) - \overline{M_s(x, z)} \right) \times \left( A(t_k) - \bar{A} \right)}{\sqrt{\sum_{k=1}^{Nst} \left( M_s(x, z, t_k) - \overline{M_s(x, z)} \right)^2} \times \sqrt{\sum_{k=1}^{Nst} \left( A(t_k) - \bar{A} \right)^2}}$$

For activation maps by SVD analysis, SVD was applied on reshaped ULM Casorati matrix M$^s$ of size ($N_x \times N_z$, $N^s_t$) (or eventually on reshaped $M$ matrix of size ($N_x \times N_z$, $N_t$)), resulting in the following decomposition based on covariance:

$$M^s(x, z, t) = \sum_{i=1}^{N^s_t} \lambda_i U_i(x, z) V_i(t),$$ with $\lambda_i$ being the singular value, $U_i$ (Nx × Nz, Nx × Nz) the spatial singular vectors and $V_i(N^s_t \times N^s_t)$ the temporal singular vectors. The singular vector $U_i$ and $V_i$ correspond to the eigenvectors of the covariance matrix M$^s$, $^tM^s$ * and $^tM^s$ * .M$^s$ where $^tM^s$ * is the transpose conjugate of M$^s$. This decomposition can be seen as a sum of images (each one corresponding to one $U_i$) independently modulated by the temporal signal $V_i$. Every pixel from the image $U_i$ behaves with the temporal fluctuations given by $V_i$.

The scalar products $p$, between the stimulation pattern signal A($t$) and every temporal singular vector, were computed to select the mode corresponding to the stimulation (Extended Data Fig. 8b–h):

$$p_i = \frac{A(t) - \bar{A}}{\sqrt{\sum_{k=1}^{N^s_t} \left( A(t_i) - \bar{A} \right)^2}} \cdot V_i(t)$$

The temporal singular vector corresponding to the brain activation was selected by detecting the highest value of the scalar products $P_i$ ($i$th vector $V_i(t)$). The spatial singular vector corresponding to the brain activation is its spatial counterpart ($i$th vector $U_i(t)$). Maps quantifying the variation in the MBs number during the period of stimulation (Figs. 4a,c,d,h,k and 5a–c, Extended Data Fig. 9a–f and Supplementary Figs. 2b and 3a) were calculated as:

$MB_{svd}(x, z)$

$= \lambda_{i=i_{stim}} \times U_{i=i_{stim}}(x, z) \times \left( \overset{t \in stim}{\int} V_{i=i_{stim}}(t)\,dt - \overset{t \in baseline}{\int} V_{i=i_{stim}}(t)\,dt \right)$

with baseline and stimulation (stim) periods having the same duration. To get the relative increase maps (Fig. 4e), we computed

$MBrelative_{svd}(x, z)$

$= \frac{\lambda_{i=i_{stim}} \times U_{i=i_{stim}}(x,z) \times \left( \int^{t \in stim} V_{i=i_{stim}}(t)dt - \int^{t \in baseline} V_{i=i_{stim}}(t)dt \right)}{\sum_{i=1}^{i<i_{stim}} \lambda_i \times U_i(x,z) \int^{t \in baseline} V_i(t)dt}$

The baseline maps (Extended Data Fig. 8a) were computed as:
$\sum_{i=1}^{i<i_{stim}} \lambda_i \times U_i(x, z) \times \overset{t \in baseline}{\int} V_i(t)\,dt$

MBs variation maps for SVD, applied to the ULM 3D temporal matrix without any pattern summation (Fig. 4f), were calculated as:

$MB_{svd}(x, z)$

$= \sum_{k=1}^{Nstim} \lambda_{i=i_{stim}} \times U_{i=i_{stim}}(x, z) \times \left( \overset{t \in stim_k}{\int} V_{i=i_{stim}}(t)\,dt - \overset{t \in baseline_k}{\int} V_{i=i_{stim}}(t)\,dt \right)$

For vascular compartment analysis, the S1BF activated area was chosen based on fUS experiments data (see 'fUS signal analysis' section). Segmentation of the vasculature was obtained by applying vesselness[60] filtering on the MB count ULM map within the activated area (2D implementation available on Mathworks file exchange)[60,61]. Pial vessels were selected manually from this segmentation. Discrimination between arterioles and venules was based on the vertical flow direction (ascending for venules and descending for arterioles). The remaining pixels, corresponding to first-order and lower-order capillaries, were labeled as intraparenchymal vessels (Fig. 2a). We have not discriminated the pial veins from pial arteries in this work as pial blood vessels were not our main interest; however, this could be performed similarly to what has been carried out for penetrating vessels. To prevent issues resulting from penetrating arterioles overlapping with venules, we only considered MBs flowing downward for the arteriolar compartment and upward for the venule compartment. There might be small vessels overlapping penetrating vessels but as the MBs flowing through them would be low compared to the number of MBs in bigger blood vessels, they would not impact the estimations. The intraparenchymal vessels compartment on the other hand does not contain any bigger blood vessels.

The compartment dynamic histograms of MB velocities (Fig. 2b) and time courses (Fig. 2c) were constructed using the same rasterization scheme as described in the section 'Temporal ULM data construction' section. Each time point was separated by a 1-s step and data from the window ($t - 2.5 : t + 2.5$) from each pattern (Nb_pattern = 40) were used. For each time point of the dynamic

histogram, a velocity histogram was constructed with every MB detection falling within the corresponding time interval and included in the vascular compartment under investigation. A 1 mm s$^{-1}$ bin was used for the histograms. The dynamic histogram was obtained by concatenating every time-point histogram. They were then normalized by the maximal bin value.

MB flux and velocity temporal signals were summed along the two spatial dimensions for each compartment. MB flow and velocity time courses were shown as mean ± s.e.m. from four time courses of ten stimulation patterns each. They were normalized by the baseline to give relative variations (Fig. 2c).

The exact same analysis was performed in the control side of the cortex (Extended Data Fig. 4).

ULM data analysis for rest versus stimulation periods. For the data used in this section, the acquisition dataset was split into two subsets, one corresponding to stimulation periods (30 s to 60 s within each repeated stimulation pattern) and one corresponding to baseline periods (0 s to 30 s within each repeated stimulation pattern). MB count and velocity maps were computed for these two subsets. They correspond to data shown in Figs. 2d–j and 3a,c, Extended Data Figs. 5 and 7 and Supplementary Table 1.

For longitudinal profile metrics (Figs. 2d–h and 3a,c), a skeletonization (MATLAB bwmorph function) was performed on the vasculature binary image (2D filtering). The skeleton corresponding to the vessel under study was selected and eventually removed from short segments to keep only the penetrating vessel centerline. For each pixel of the centerline (corresponding to each cortical depth of the vessel), the flow direction was measured (computed as $\tan^{-1}\left(\frac{Vx}{Vz}\right)$) and an 80-μm segment normal to the flow direction and centered on the centerline was chosen to compute the MB count and velocity profiles. The following metrics were extracted for each depth: max MB count, velocity at max MB count, diameter. The diameter was defined as the width of the vessel at a threshold set by the half maximum of the rest profile. A smoothing filter (window of 200 μm) was applied on max MB count and velocity values along the depth dimension. The same was performed on the diameter value using a median filtering (presence of outliers due to bifurcations).

For transversal profiles (Fig. 2i,j, Extended Data Fig. 7b,c and Supplementary Table 1), a segment crossing the vessel under study at the desired depth was chosen. The flow direction (defined as $\tan^{-1}\left(\frac{Vx}{Vz}\right)$) was computed at the pixel corresponding to the maximum MB count (center of the vessel). MB count and velocity profiles were computed by averaging on a 50-μm wide slice of an 80-μm segment normal to the flow direction and centered on the pixel with maximal MB count. Profiles were smoothed (50 μm) before extracting the following metrics: max MB count, velocity at max MB count and diameter (same definition as above).

For perfusion or drainage area index, a segment crossing the cortical vessel under study was set near the pia matter (black segment in Extended Data Fig. 5a). Every MB track that passed through the selected segment was retained. Examples of different arterioles and venules (Extended Data Fig. 5b) highlight the capacity of ULM to detect first orders of branching. This confirms the tree-like structure of penetrating vessels with varying penetration depths (as described previously[64]). The positions they describe were rounded to a 6.5-μm pixel size. The area perfused by the vessel was defined by the number of pixels where at least one track position had been counted, multiplied by the pixel area (Fig. 2d–g, Extended Data Fig. 7b,c and Supplementary Table 1).

For vessel selection and metrics computation for statistical analysis (Fig. 2i,j and Extended Data Fig. 7 and Supplementary Table 1), we distinguished four categories of blood vessels: control arterioles, control venules, activated barrel arterioles and activated barrel venules. Activated barrel vessels were chosen randomly within the activated area (Extended Data Fig. 7a) defined by the fUS experiment (see 'fUS signal analysis' section). Control vessels were chosen in the contralateral cortex from the activated area. A total of 20 control arterioles and activated barrel arterioles (out of around 65 potential arterioles in total in the activated areas) and 18 control venules and activated barrel venules (out of around 38 potential venules in total in the activated areas) were chosen in four different animals (at least three vessels per animal for each category). Transversal profiles were computed (following the method described above) for each vessel for both rest and stimulation periods at two different depths: less than 400 μm from pial vessels and more than 600 μm from pial vessels. For the second depth, only 15 profiles could be computed for activated barrel as three venules did not penetrate deeply enough into the cortex. Max MB count, velocity and diameter were extracted for each of the profiles (see 'Rest and stim longitudinal and transversal profiles' section). The perfused area was also measured for each of the 76 vessels for both rest and stimulation periods.

For dilatation and constriction maps, a segmentation of the vasculature was obtained by applying a vesselness[60] filtering on the MB count ULM map computed both on the rest period and on the stimulation periods (2D implementation available on Mathworks file exchange)[60,61]. The two binary maps were then subtracted.

**Statistics and reproducibility.** Ten animals were used in this study. Supplementary Table 2 summarizes which animal was used for each figure or result.

In Extended Data Fig. 3, we used $n = 4$ rats for whisker stimulations (20 stimuli repetitions) and $n = 3$ rats for visual stimulations (20 stimuli repetitions). The ROI were selected using fUS imaging experiments, as described in 'Functional ultrasound activation maps and ROI selection' and were used to extract one temporal response for each ROI and for each animal. In one of the four experiments, no thalamus activation could be seen in the fUS experiment. ULM maps were also registered in the Paxinos Atlas referential[52]. The temporal responses shown in Extended Data Fig. 3 are shown as mean ± s.e.m. for those $n = 4$ (except for thalamus, $n = 3$) rats.

In Fig. 2c, 40 whisker stimulations (two experiments of 20 repetitions) were used on the same rat and the same imaging plane and $n = 4$ temporal responses based on 10 stimuli each were computed. The data in Fig. 2c (and Extended Data Fig. 4 and Supplementary Fig. 4) are represented as mean ± s.e.m. for those $n = 4$ responses.

In Fig. 2i,j and Extended Data Fig. 7, we used whisker stimulation experiments (20 stimulations) on $n = 4$ rats to select $n = 20$ activated barrel arterioles, $n = 20$ control arterioles, $n = 18$ activated barrel venules, $n = 18$ control venules (at least three vessels per rat for each category) (see section 'Vessel selection and metrics computation for statistical analysis'). For each vessels' category we computed for every metric (max MB count, velocity, diameter, perfusion or drainage area): the rest value (given as mean ± s.e.m.) and the variation during stim periods normalized to baseline (given as mean ± s.e.m.) at two different depths (<400 μm and >600 μm from pial vessels). As mentioned in 'Vessel selection and metrics computation for statistical analysis', for the second depth, only 15 profiles could be computed for barrel venules, as three venules did not penetrate deeply enough into the cortex. As this relative variation did not follow a normal distribution for some of the metrics (tested with an Anderson–Darling test for normality), we chose a nonparametric test. A two-sided Wilcoxon signed-rank test (null hypothesis, distribution with a zero median) was applied to the relative variation for every metric. Results are given as $P$ values and are summarized in Supplementary Table 1 and Extended Data Fig. 7b box plots. The difference between the relative increases between the two depths of MB count, speed and diameter was tested using a paired, two-sided Wilcoxon signed-rank test (null hypothesis, difference between the two variables comes from a distribution with zero median). Results are given as $P$ values.

All the examples shown for the typical experiments in Figs. 1b,g,h, 3b and 4a–e, Extended Data Figs. 1, 2d,f, 3a,b, 5b,c, 7a,c, 8a, 9 and 10e–j and Supplementary Figs. 1, 4 and 5 were repeated $n = 4$ times for whisker experiments and $n = 3$ times for visual experiments. Some experiments (the remaining examples shown) were performed once as they are only to be considered as perspectives of the work presented in this manuscript (such as short stimulations or transcranial imaging, respectively, in Supplementary Figs. 2 and 3).

**Microflow rendering for video representation.** *Animated flow rendering.* Computed-generated imagery animation (Supplementary Videos 1 and 2) was created using the software Houdini 18.5.563 (SideFX) to visualize the functional information extracted from dynamic bubble positions and tracking information.

*MB count variation (Supplementary Video 1).* The MB flux variation (MBs per pixel per second) and mean velocity vector ($Vx$, $Vz$) (corresponding to MB$_F$ ($x$, $z$, $t$) and $V^s(x, z, t)$, section 'Temporal ULM data construction') acquired from whisker stimulations pattern (70 blocks of 1 s each consisted of 30 s rest, 30 s stimulation and 10 s rest repeated 20 times and summed) were imported into Houdini using a custom Python node (Supplementary Video 1). To produce smooth animation with 420 frames (played at 24 f.p.s.) using the initial 70 frames, data were linearly interpolated using a 'retime' node.

Next, inside a particle operator (POP) network, particles life was set to 0.1 s (2.4 frames) and their position was updated for each frame based on the imported velocity vector field using the node 'POP advect by volumes'. Finally, flux increase, in respect to baseline, was highlighted by replicating particles proportionally to the value of MBs per pixel per second, using the 'POP replicate' node with a maximum of four emitted particles per second having a lifetime of 0.5 s (12 frames). And to better emphasize the flux increase, particle color was converted from RGB to hue, saturation and value color space and the color value (intensity) was modified proportionally to the value of MBs per pixel per second.

*MB trajectories (Supplementary Video 2).* MBs dataset point gathering track ID, position ($x$, $z$) and velocity for activated barrel venule or arteriole and control venule or arteriole were imported into Houdini. MB trajectories were reconstructed using an 'add' node using track ID attribute to create a solid line and then animated using a 'carve' node (Supplementary Video 2).

**Reporting summary.** Further information on research design is available in the Nature Research Reporting Summary linked to this article.

## Data availability
ULM data for data analysis[50] are provided on the Zenodo repository website at https://doi.org/10.5281/zenodo.6109803.

## Code availability

Data analyses were performed on MATLAB R2020b (MathWorks). Home-made MATLAB codes were used for the ULM algorithms: SVD filtering of tissue signal (according to Demené et al.[54]) and localization of the MBs. The ULM algorithm includes vesselness filtering available on Mathworks file exchange (2D implementation available on Mathworks file exchange)[60,61] and a tracking algorithm, simpletracker.m available on Mathworks Tinevez, wrapping the MATLAB munkres algorithm implementation of Cao, 2009. Home-made MATLAB codes were used for the ULM data analysis and the statistical analysis. Super-resolution videos were obtained using a 3D software for visualization (Houdini, 17.5.360, SideFX). MATLAB codes for the reading of ULM data[50] are provided on the Zenodo repository website at: https://doi.org/10.5281/zenodo.6109803. Low-level acquisition and processing codes of the raw data used for the collection of ULM data are protected by INSERM and can only be shared upon request, with the written agreement of INSERM.

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

## Acknowledgements

This research was funded by the AXA Research Fund (AXA Chair New Hopes in Medical Imaging with Ultrasound) and the INSERM Accelerator of Technological Research in Biomedical Ultrasound. The authors thank the NVIDIA Corporation for offering a GPU board through the NVIDIA GPU Grant program. The NVIDIA company did not have any influence on the research.

## Author contributions

N.R., C.D., S.P. and M.T. designed the experiments. N.I.-R. and S.P. performed the surgeries. N.R. and S.P. performed the experiments and acquired the ultrasound data. N.R., C.D. and M.T. analyzed the data. A.D. designed the videos. N.R. and M.T. wrote the paper with additional contributions of S.P. and C.D.

## Competing interests

M.T. is a co-founder and shareholder of the Iconeus company, which commercializes ultrasound neuroimaging scanners. M.T. is a co-inventor of the patent WO2012080614A1 filed on 16 December 2010 and licensed to the Iconeus company. All other authors declare no competing interests.

## Additional information

**Extended data** is available for this paper at https://doi.org/10.1038/s41592-022-01549-5.

**Correspondence and requests for materials** should be addressed to Mickael Tanter.

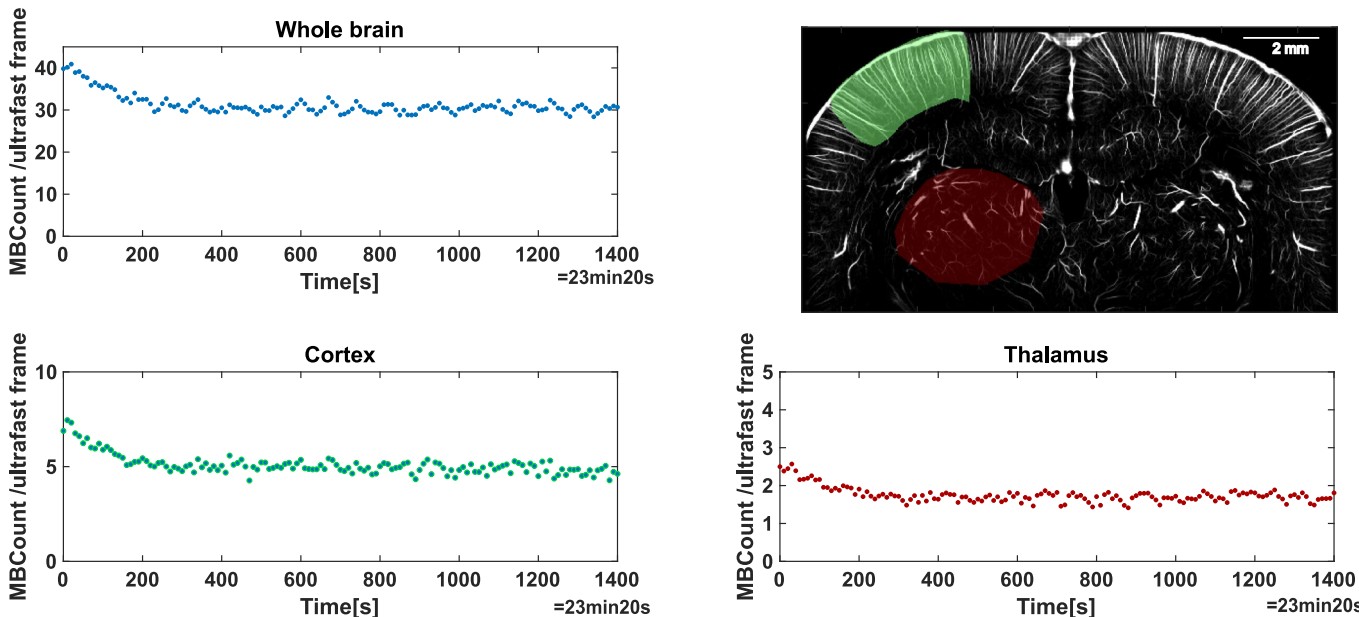

**Extended Data Fig. 1 | Microbubble injection profile.** MB detection count per ultrafast frame count for the whole brain, cortical and thalamic regions during continuous perfusion of MBs over the whole acquisition (23 minutes). The ULM map displays the whole brain (no highlighting), cortical area (green) and thalamic area (red) used for MB detection count, N = 7 experiments.

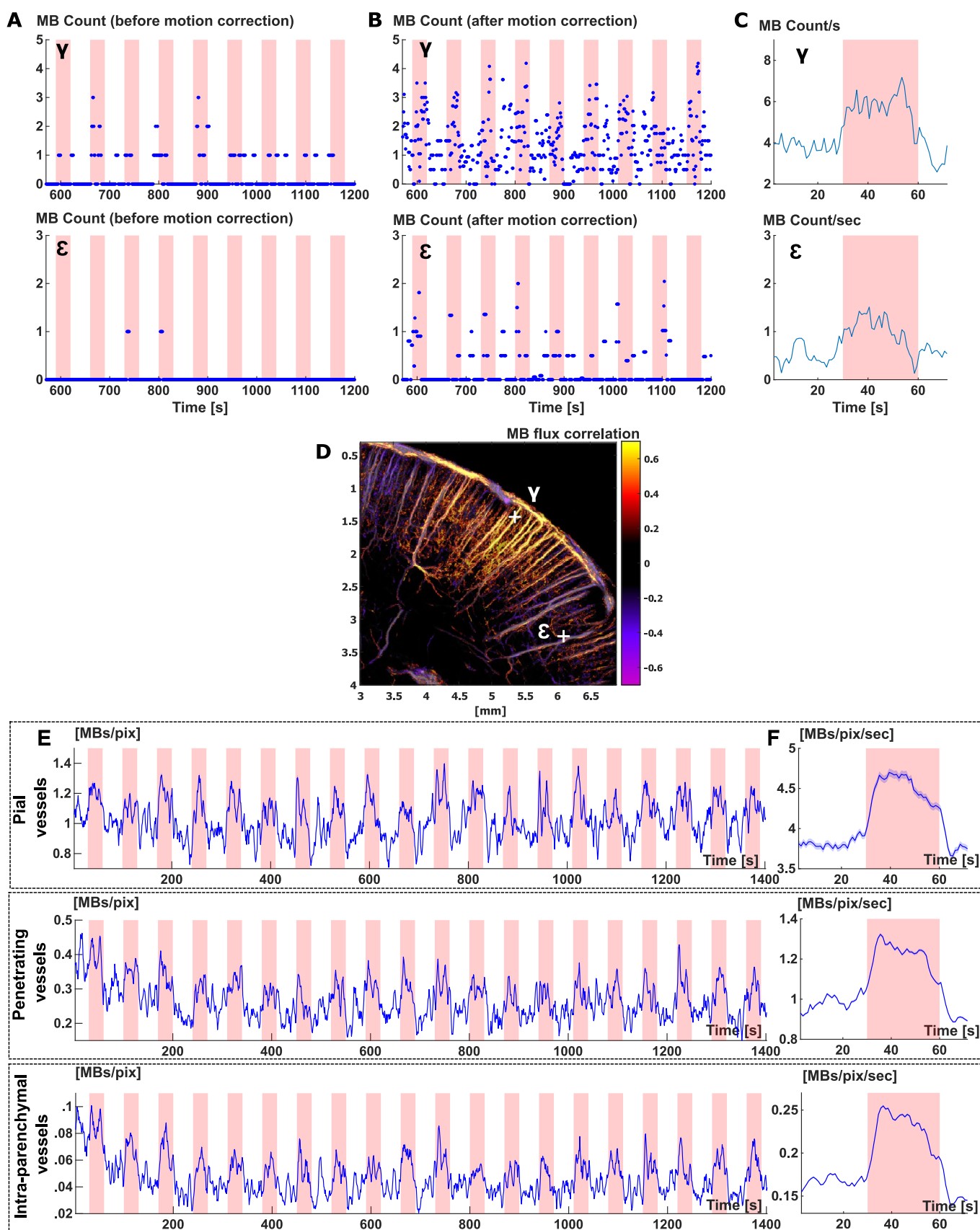

**Extended Data Fig. 2 | See next page for caption.**

**Extended Data Fig. 2 | Temporal profiles of microbubble detection in typical pixels representative of pial vessels, penetrating vessels and 1$^{st}$ order branching capillaries.** (**A**) Time courses of MB count for a pixel in an arteriole (top, γ in D) and smaller blood vessel (bottom,1$^{st}$ order branching after descending arteriole, ε in D), illustrated in D. The red shaded intervals correspond to stimulation periods. (**B**) Time courses of MB count for the same pixels after spatial registration. (**C**) Time courses of MB flux for the same pixels after pattern summation. (**D**) Position of pixel γ and ε on a functional correlation map. (**E**) Time courses of MB count after spatial registration averaged over three different vascular compartments (pial vessels, penetrating vessels and 1st order branching capillaries). (**F**) Time courses of MB flux averaged over the same three vascular compartments as in (E), after pattern summation. (A-F) N = 7 experiments.

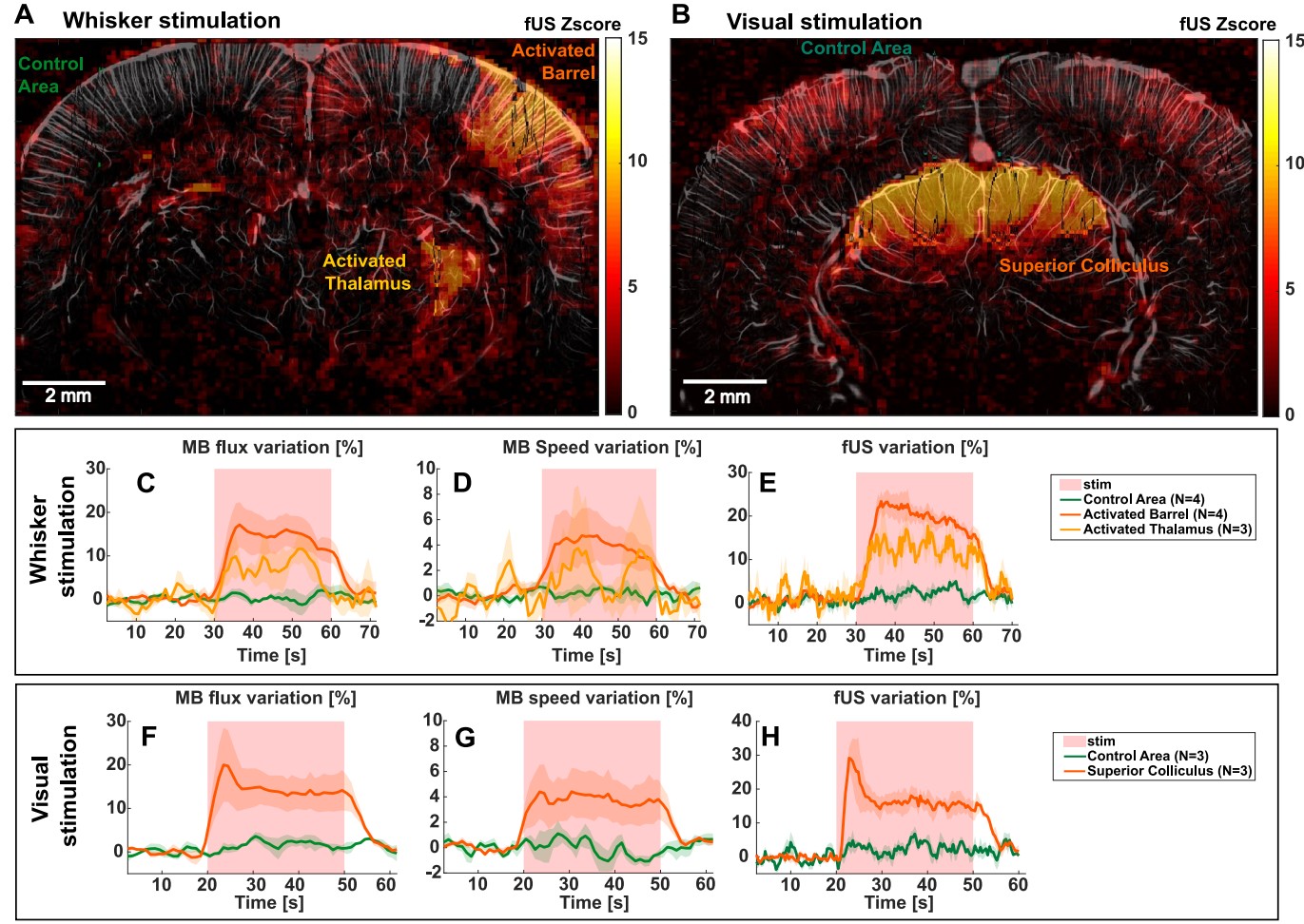

**Extended Data Fig. 3 | Inter-animal reproducibility: MB flux and velocity temporal responses during functional hyperemia. (A-B)** fUS imaging: examples of activated barrel (A), thalamus (A), superior colliculus (B) as well as control areas chosen for temporal responses extraction. They are shown for one whisker and one visual task-evoked experiments on a ULM MB count map overlaid with a fUS activation map. Activated zones are chosen using the Z-score with a threshold at $p < 0.05$ after Bonferroni correction **(C-H)** Quantitative fULM imaging: MB flux and velocity temporal variations in the barrel cortex (N = 4) and thalamus (N = 3) for whiskers stimulation **(C-D)** and superior colliculus for visual stimulation (N = 3 rats, **F-G**) respectively. Power Doppler signals are shown for the corresponding fUS experiments for comparison (**E-H**). Data are presented as mean ± SE.

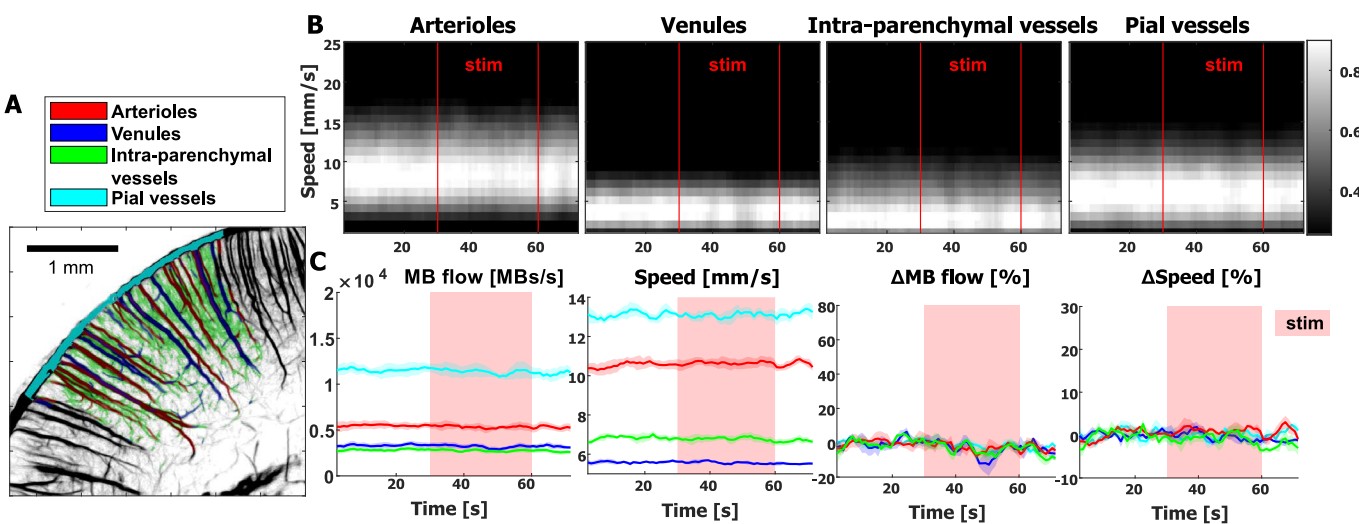

**Extended Data Fig. 4 | Vascular compartment analysis in control area.** (**A**) The control area (located in the cortex ipsilateral to the whisker stimulation) is subdivided in 4 different vascular compartments (penetrating arterioles, venules, pial vessels and intraparenchymal vessels) based on the super-resolved ULM maps. This segmentation is shown in colors overlaid on the ULM map. (**B**) Velocity spectrograms showing the velocity distribution in the different blood vessels defined in (A) and how it is modified during whiskers stimulations (N = 40 stimuli). (**C**) Mean MB flow and speed (±SE) from N = 4 different time courses obtained on 10 stimulations each, either expressed as absolute value for each type of blood vessels (two left panels), or as relative to baseline (two right panels).

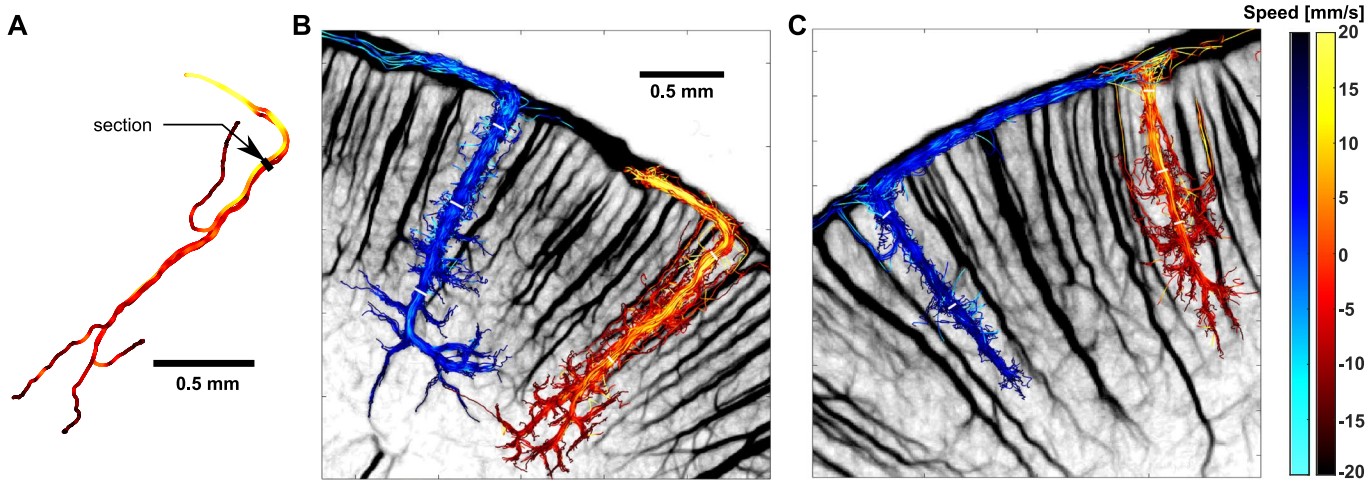

**Extended Data Fig. 5 | Lagrangian description of blood vessels.** (**A**) Drawing of a vessel segment based on the ULM map. Four MB tracks crossing the black segment. The speed of the MB is color-coded (see colorbar on the right). (**B**) Example of an arteriole and a venule penetrating deep in the cortex, the selected MB tracks are the one crossing the white segments. (**C**) Same as in (B) but for a shorter venule and arteriole. (A-C) N = 4 experiments.

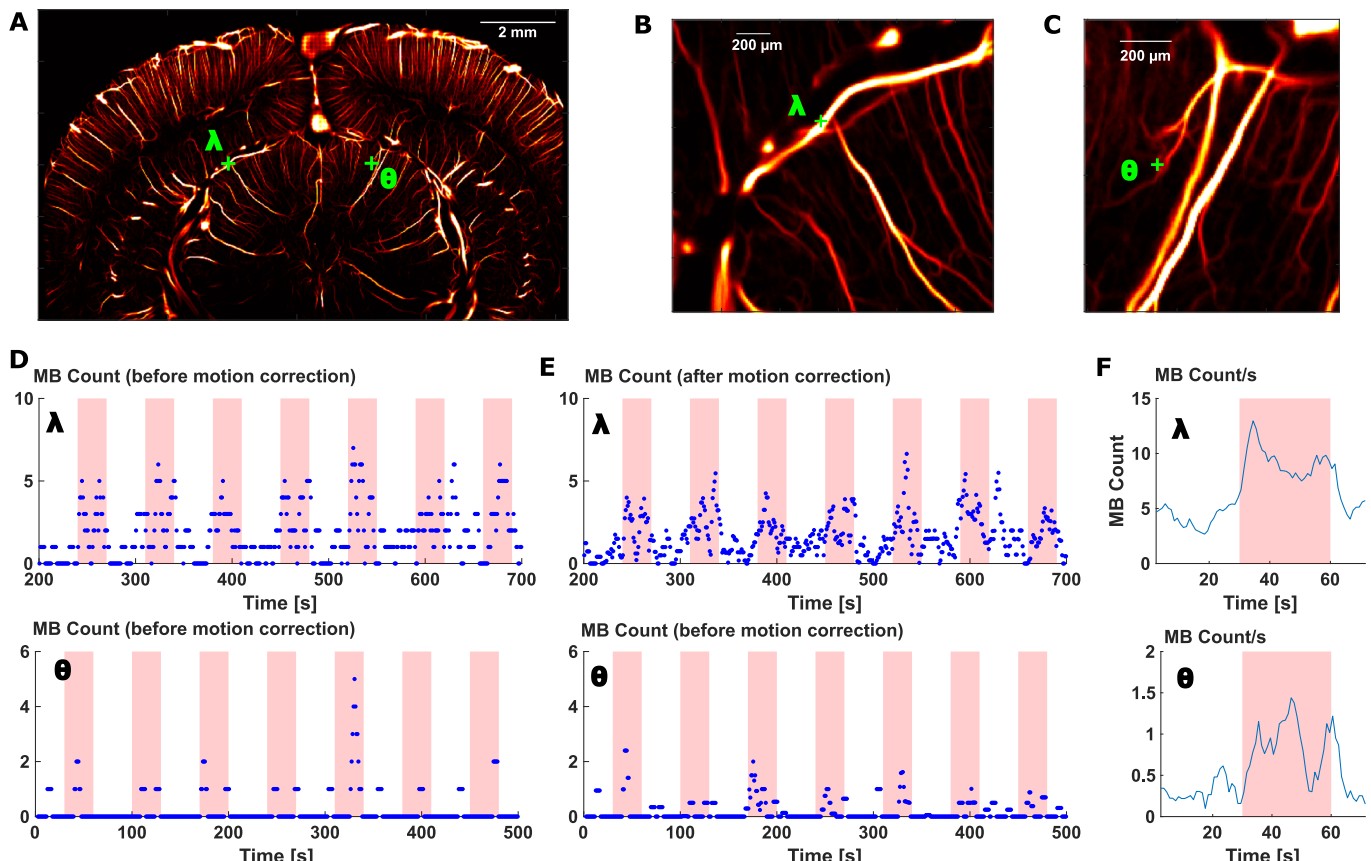

**Extended Data Fig. 6 | Temporal profile of the MB flow in pixels from deep-seated vessels of the superior colliculus during visual stimulation.** (**A**) Location of pixels λ (in a 60 μm diameter vessel) and θ (in a 15 μm diameter vessel) in the superior colliculus. (**B-C**) Zoomed regions of interest from image (A). (**D-E**) Temporal profile of MB count in pixels λ and θ before and after motion correction during the successive visual stimuli. (**F**) Temporal profile of MB count in pixels λ and θ after pattern summation. (A-F) N=3 experiments.

A

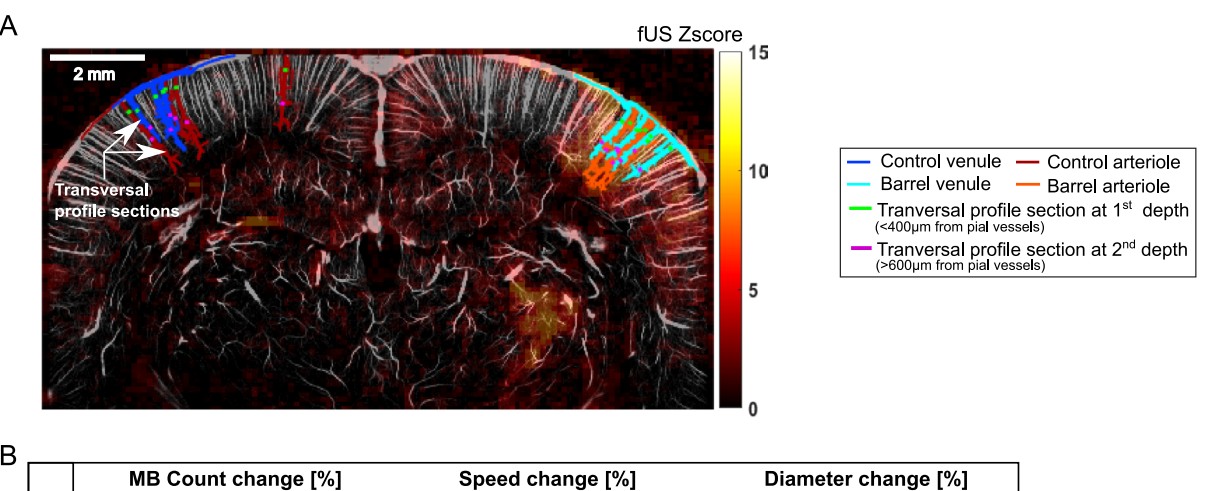

fUS Zscore

— Control venule — Control arteriole
— Barrel venule — Barrel arteriole
— Tranversal profile section at 1st depth (<400μm from pial vessels)
— Tranversal profile section at 2nd depth (>600μm from pial vessels)

Transversal profile sections

B

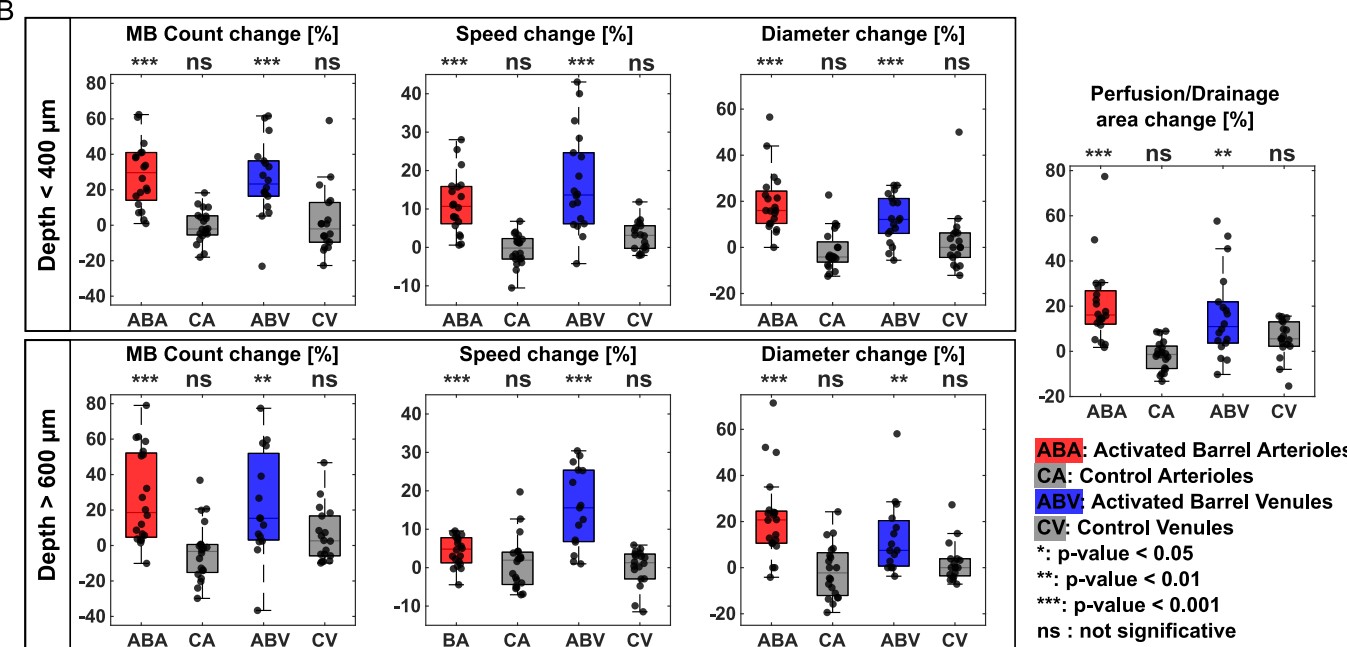

ABA: Activated Barrel Arterioles
CA: Control Arterioles
ABV: Activated Barrel Venules
CV: Control Venules
*: p-value < 0.05
**: p-value < 0.01
***: p-value < 0.001
ns : not significative

C

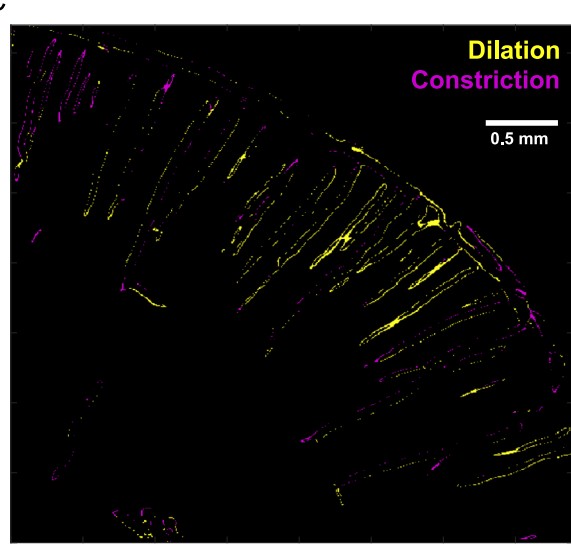

Dilation
Constriction

0.5 mm

**Extended Data Fig. 7 | See next page for caption.**

**Extended Data Fig. 7 | Detailed statistical analysis of fULM signals in selected vessels during whisker stimulations.** (**A**) Illustration of the selection of activated barrel and contralateral cortex blood vessels (arterioles and venules in the activated barrel cortex and contralateral cortex for controls). Profiles were measured at two depths: <400 µm (green marks) and >600 µm (magenta marks) from pial vessels. (**B**) Boxplots corresponding to rest value and variation relative to rest (mean ± SE), p-value for two-sided Wilcoxon signed rank test on this variation (null hypothesis: distribution with median equal to zero) for MB Count, speed, diameter and perfusion for the different categories of blood vessels. The number of animals for depth 1 are N = 20 (ABA and CA) and N = 18 (ABV and CV). The number of animals for depth 2 are N = 20 (ABA and CA), N = 18 (CV), N = 15 (ABV). The number of animals for perfusion and drainage area measurements are N = 20 (ABA and CA) and N = 18 (ABV and CV). The central mark indicates the median, and the bottom and top edges of the box indicate the 25th and 75th percentiles, respectively. The whiskers extend to the most extreme data points not considered outliers. Scatter plots of the data used for the boxplot are overlaid on each boxplot. All results of the Wilcoxon test are detailed in Supplementary Table 1 and summarized on top of each boxplot. (**C**) Dilatation and Constriction map. (**A-C**) N = 4 experiments.

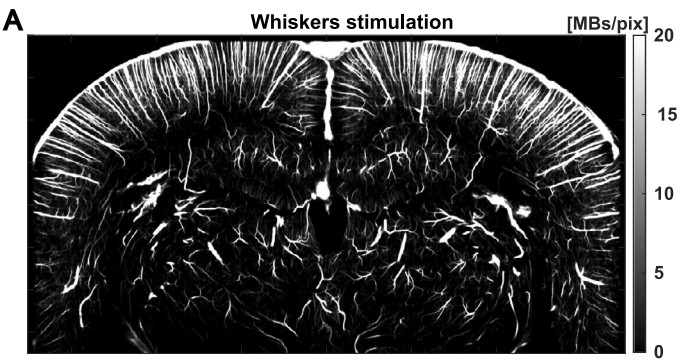

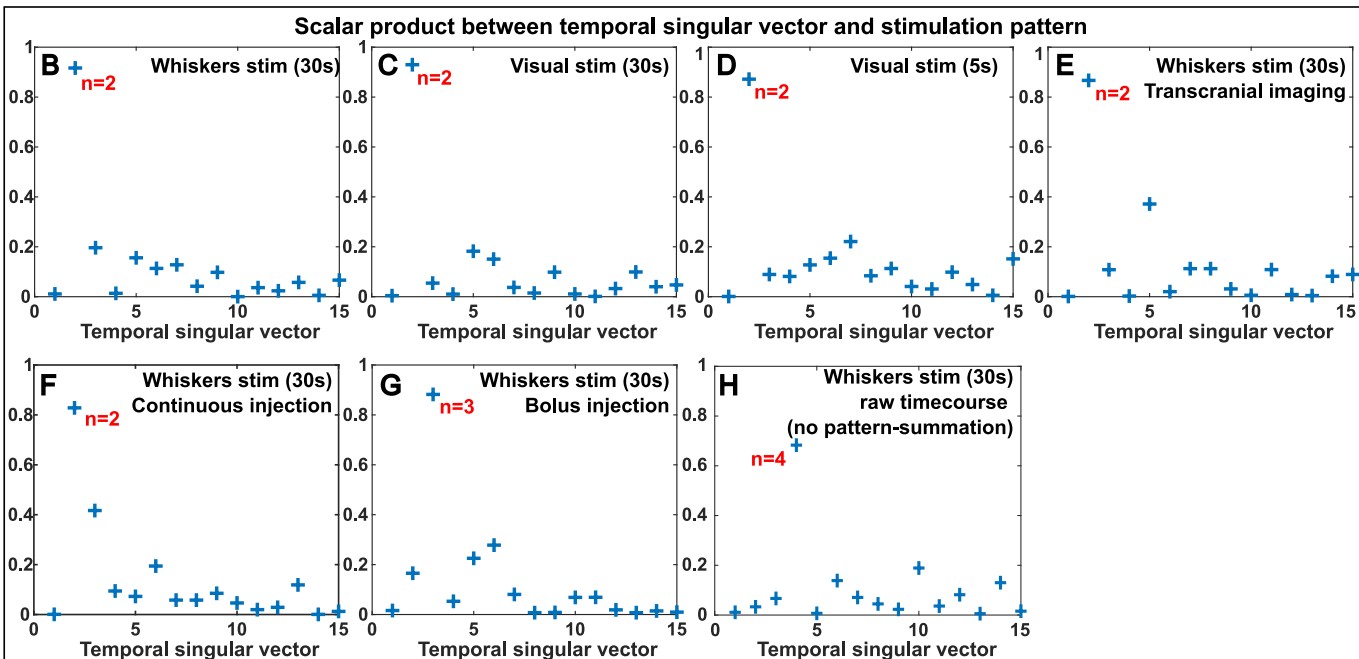

**Extended Data Fig. 8 | Extraction of task-evoked signature using SVD analysis. (A)** Baseline signal is provided by the first spatial singular vector after SVD decomposition on whisker stimulation. (**B-H**) Scalar product of the first 15 temporal singular vectors with the stimulation pattern: automatic selection of the singular vectors containing task-evoked brain activity during SVD analysis. (A-H) Single micrograph.

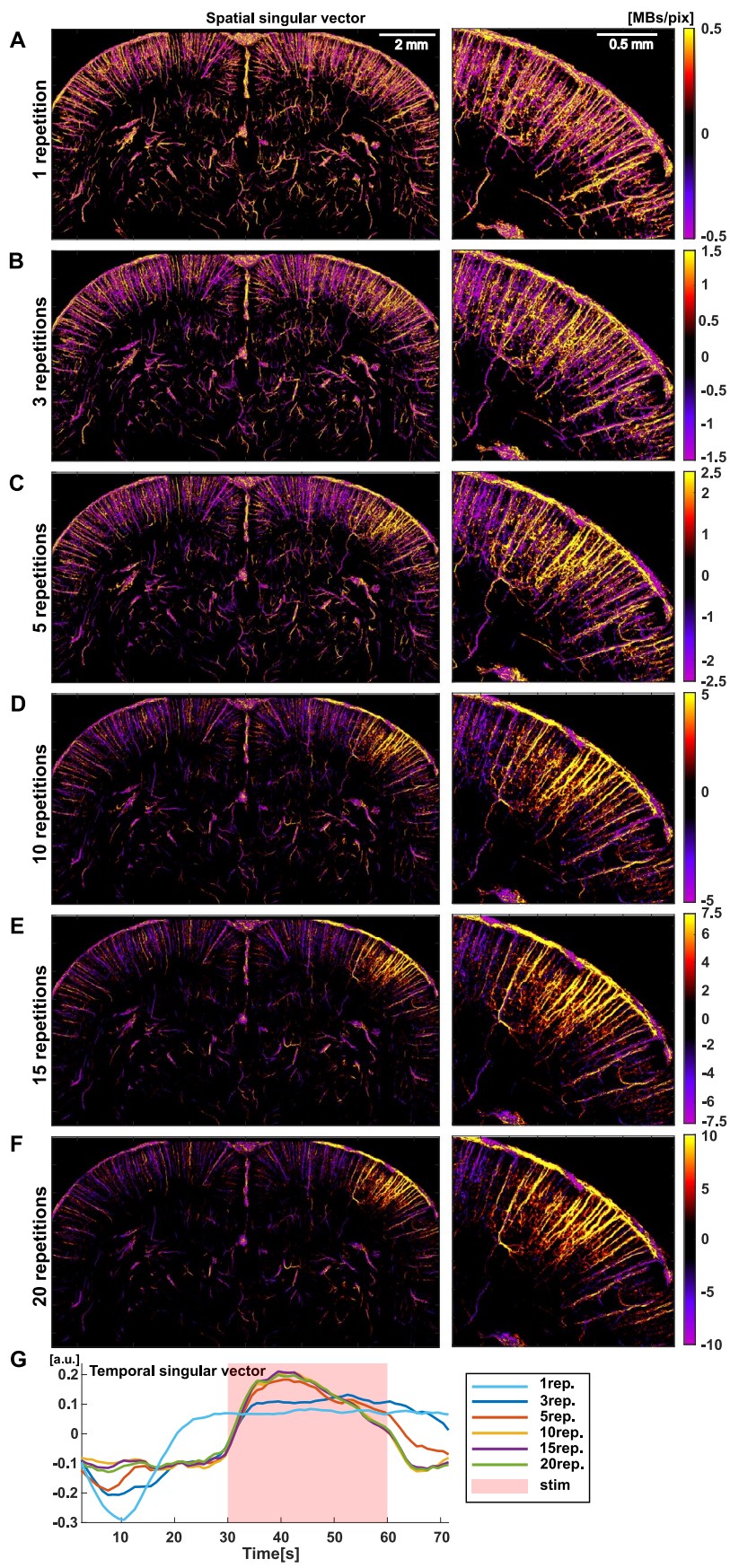

**Extended Data Fig. 9 | See next page for caption.**

**Extended Data Fig. 9 | Functional ULM imaging by SVD analysis for an increasing number of stimulation pattern repetitions.** SVD analysis performed on the MB flux signal. (**A**-**F**) The different panels show the spatial singular vectors corresponding to the stimulation (left: whole brain; right: zoomed area in the activated barrel cortex) for an increasing number of pattern repetition, from N = 1 (**A**) to N = 20 (**F**). (**G**) Time courses of the corresponding temporal singular vector. (A-G) N = 7 experiments.

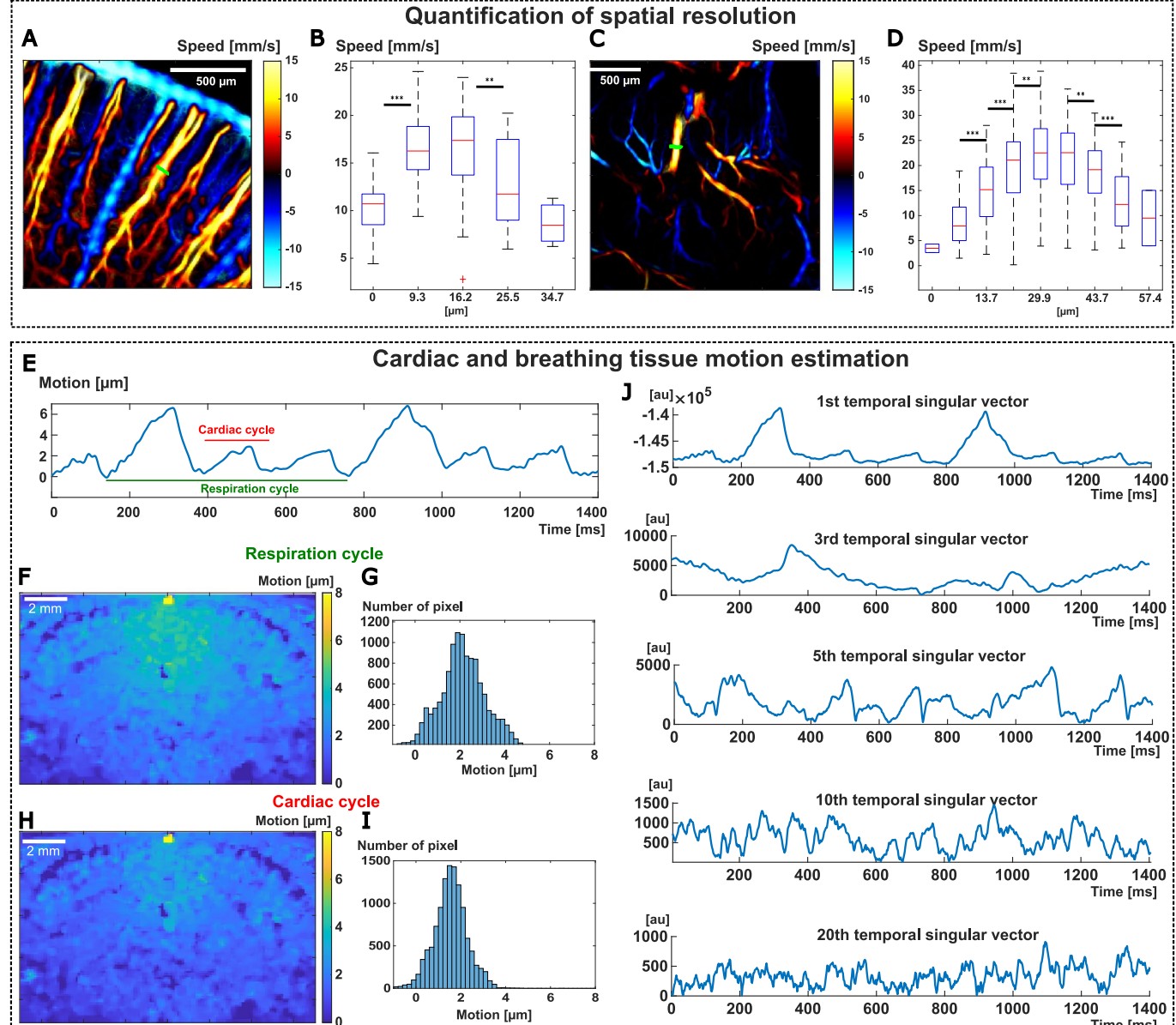

**Extended Data Fig. 10 | Quantification of fULM spatial resolution, cardiac and respiratory motion. (A-C) Quantification of the spatial resolution**. Signed velocity map of the rat brain vasculature showing two vessels profiles (green) chosen for the quantification in the cortex (A-B) and deep thalamic region (C-D). (B-D) Velocity profiles for each profile displayed in (A-C). Bubbles velocities from one time point of the pattern-averaged data are gathered in bins corresponding to each [6.875 6.25] µm² pixel. The central mark of the boxplot indicates the median, and the bottom and top edges of the box indicate the 25th and 75th percentiles, respectively. The p-values for the a two-sampled t-test between each consecutive bin are indicated as: *P < 0.05,**P < 0.01, ***P < 0.001. Samples number for each bin from left to right: (B): [11, 27,26,17,3]. (D) [2, 47,96,179,223,194,135, 50, 2]. **(E-J) Estimation of cardiac and breathing tissue motion. (E)** Spatially averaged tissue motion along the z-axis exhibits cardiac and breathing pulsatility (respiration cycle: ~620 ms / ~97bpm; cardiac cycle: ~180 ms / ~330bpm). (**F-G**) Mean tissue motion over a respiratory cycle evaluated in each pixel represented in a map (B) and in a histogram (C). (**H-I**) Mean tissue motion over a cardiac cycle evaluated in each pixel represented in a map (D) and in a histogram (E). (**J**) Temporal singular vector (i = 1,3,5,10,20) multiplied by each respective singular value of the singular value decomposition of the same raw IQ data used in (E). (E-J) N = 4 experiments.

# Reporting Summary

## Statistics

For all statistical analyses, confirm that the following items are present in the figure legend, table legend, main text, or Methods section.

| n/a | Confirmed | |
|---|---|---|
| ☐ | ☒ | The exact sample size (*n*) for each experimental group/condition, given as a discrete number and unit of measurement |
| ☐ | ☒ | A statement on whether measurements were taken from distinct samples or whether the same sample was measured repeatedly |
| ☐ | ☒ | The statistical test(s) used AND whether they are one- or two-sided *Only common tests should be described solely by name; describe more complex techniques in the Methods section.* |
| ☐ | ☒ | A description of all covariates tested |
| ☐ | ☒ | A description of any assumptions or corrections, such as tests of normality and adjustment for multiple comparisons |
| ☐ | ☒ | A full description of the statistical parameters including central tendency (e.g. means) or other basic estimates (e.g. regression coefficient) AND variation (e.g. standard deviation) or associated estimates of uncertainty (e.g. confidence intervals) |
| ☐ | ☒ | For null hypothesis testing, the test statistic (e.g. *F*, *t*, *r*) with confidence intervals, effect sizes, degrees of freedom and *P* value noted *Give P values as exact values whenever suitable.* |
| ☒ | ☐ | For Bayesian analysis, information on the choice of priors and Markov chain Monte Carlo settings |
| ☒ | ☐ | For hierarchical and complex designs, identification of the appropriate level for tests and full reporting of outcomes |
| ☒ | ☐ | Estimates of effect sizes (e.g. Cohen's *d*, Pearson's *r*), indicating how they were calculated |

*Our web collection on statistics for biologists contains articles on many of the points above.*

## Software and code

Policy information about availability of computer code

| Data collection | Raw Ultrasound data was acquired using a linear ultrasound probe driven by a prototype ultrasonic ultrafast neuroimager (Iconeus, France) and the with Neuroscan live acquisition software (version 1.3, Iconeus, Paris, France and Inserm Accelerator of Technological Research in Biomedical Ultrasound, Paris, France). |
|---|---|
| Data analysis | Data analysis were performed on Matlab R2020b (MathWorks, Cambridge, MA, USA). Home-made Matlab codes were used for the ULM algorithms: SVD filtering of tissue signal (according to Demené, C. et al. Spatiotemporal Clutter Filtering of Ultrafast Ultrasound Data Highly Increases Doppler and fUltrasound Sensitivity. IEEE Transactions on Medical Imaging 34, 2271–2285 (2015) ), localization of the microbubbles. The ULM algorithm includes a vesselness filtering available on Mathworks file exchange (2D implementation available on Mathworks file exchange, ©Dirk-Jan Kroon2009, and © Tim Jerman, 2017) and a tracking algorith, simpletracker.m available on Mathworks ©Jean-Yves Tinevez, wrapping matlab munkres algorithm implementation of ©Yi Cao 2009.Home-made Matlab codes were used for the ULM data analysis and the statistical analysis. Superresolution movies were obtained using a 3D software for visualization (Houdini, 17.5.360, SideFX, Toronto, Canada). Matlab codes for the reading of ULM data are provided on the Zenodo repository website at: 10.5281/zenodo.6109803 Low level acquisition and Processing codes of the raw data used for the collection of ULM data are protected by INSERM and can only be shared upon request, with the written agreement of INSERM. |

For manuscripts utilizing custom algorithms or software that are central to the research but not yet described in published literature, software must be made available to editors and reviewers. We strongly encourage code deposition in a community repository (e.g. GitHub). See the Nature Portfolio guidelines for submitting code & software for further information.

## Data

Policy information about availability of data

All manuscripts must include a data availability statement. This statement should provide the following information, where applicable:

- Accession codes, unique identifiers, or web links for publicly available datasets
- A description of any restrictions on data availability
- For clinical datasets or third party data, please ensure that the statement adheres to our policy

ULM Data for data analysis are available on the zenodo repository website at: 10.5281/zenodo.6109803

# Field-specific reporting

Please select the one below that is the best fit for your research. If you are not sure, read the appropriate sections before making your selection.

☒ Life sciences      ☐ Behavioural & social sciences      ☐ Ecological, evolutionary & environmental sciences

For a reference copy of the document with all sections, see nature.com/documents/nr-reporting-summary-flat.pdf

# Life sciences study design

All studies must disclose on these points even when the disclosure is negative.

| | |
|---|---|
| Sample size | No sample size calculation was performed as the proof of concept of of functional Ultrasound Localization Microscopy was successful in all N=10 animals. Sample size was chosen by considering 100% success rate in detecting and mapping a functional brain activity after N>=3 animals was sufficient. N=4 animals were used for whiskers stimualtions. N=3 animals were used for visual stimulations. N=3 additional animals were used as examples for different experimental conditions testing. |
| Data exclusions | Preliminary experiments (N=4 animals) were used for protocol optimization (mainly microbubble injection optimization). All animals involved after that were successful and used in the study, no data exclusion. |
| Replication | For this proof of concept experiments, all attempts at replication were successful. Processing steps were performed the same way for all animals, following what is described in the manuscript. |
| Randomization | We only used naive animals. Among them, the animals were chosen randomly. No group comparison was performed in this proof of concept publication. |
| Blinding | Blinding was not relevant as ou study consists in a technical proof of concept study, and does not involve comparison between groups. |

# Reporting for specific materials, systems and methods

We require information from authors about some types of materials, experimental systems and methods used in many studies. Here, indicate whether each material, system or method listed is relevant to your study. If you are not sure if a list item applies to your research, read the appropriate section before selecting a response.

### Materials & experimental systems

| n/a | Involved in the study |
|---|---|
| ☒ | ☐ Antibodies |
| ☒ | ☐ Eukaryotic cell lines |
| ☒ | ☐ Palaeontology and archaeology |
| ☐ | ☒ Animals and other organisms |
| ☒ | ☐ Human research participants |
| ☒ | ☐ Clinical data |
| ☒ | ☐ Dual use research of concern |

### Methods

| n/a | Involved in the study |
|---|---|
| ☒ | ☐ ChIP-seq |
| ☒ | ☐ Flow cytometry |
| ☒ | ☐ MRI-based neuroimaging |

## Animals and other organisms

Policy information about studies involving animals; ARRIVE guidelines recommended for reporting animal research

| | |
|---|---|
| Laboratory animals | Experiments were performed on N=10 male Sprague–Dawley rats (Janvier Labs; Le Genest St Isle, France), weighing 200-300g (Age 7-9 weeks) |
| Wild animals | The study did not involve wild animals. |

| Field-collected samples | The study did not involve samples collected from the field. |
| --- | --- |
| Ethics oversight | All experiments were performed in agreement with the European Community Council Directive of September 22, 2010 (010/63/UE) and the local ethics committee (Comité d'éthique en matière d'expérimentation animale N°59, 'Paris Centre et Sud', project #2017-23) |

Note that full information on the approval of the study protocol must also be provided in the manuscript.

