## [Peer Review File · Nature Methods]

Peer Review Information

Manuscript Title: Functional Ultrasound Localization Microscopy reveals brain-wide neurovascular activity on a microscopic scale

Corresponding author name(s): Mickael Tanter

Editorial Notes:

Redactions – transferred manuscripts (mention of previous referee reports from elsewhere)	This manuscript has been previously reviewed at another journal. This document only contains reviewer comments, rebuttal and decision letters for versions considered at Nature Methods. Mentions of prior referee reports have been redacted
Redactions – transferred manuscripts (mention of the other journal)	This manuscript has been previously reviewed at another journal. This document only contains reviewer comments, rebuttal and decision letters for versions considered at Nature Methods. Mentions of the other journal have been redacted.
Redactions – unpublished data	Parts of this Peer Review File have been redacted as indicated to maintain the confidentiality of unpublished data.
Redactions – published data	Parts of this Peer Review File have been redacted as indicated to remove third-party material.

Reviewer Comments & Decisions:

Decision Letter, initial version:

Dear Mickael,

Thank you for transferring your Article, "Functional Ultrasound Localization Microscopy reveals brain-wide neurovascular activity on a microscopic scale", to Nature Methods. Before we can assess the manuscript in detail, could you provide a rebuttal letter to the comments from reviewers received at [Redacted]? Please also revise the manuscript accordingly.

[Redacted] This URL links to your confidential home page and associated information about manuscripts you may have submitted, or that you are reviewing for us. If you wish to forward this email to co-authors, please delete the link to your homepage.

We hope to receive your revised paper within 4 weeks. If you cannot send it within this time, please let us know. In this event, we will still be happy to reconsider your paper at a later date so long as nothing similar has been accepted for publication at Nature Methods or published elsewhere.

OPEN SCIENCE REQUIREMENTS

REPORTING SUMMARY AND EDITORIAL POLICY CHECKLISTS

Please note that these forms are dynamic ‘smart pdfs’ and must therefore be downloaded and completed in Adobe Reader. We will then flatten them for ease of use by the reviewers. If you would like to reference the guidance text as you complete the template, please access these flattened versions at <http://www.nature.com/authors/policies/availability.html>.

DATA AVAILABILITY

All novel DNA and RNA sequencing data, protein sequences, genetic polymorphisms, linked genotype and phenotype data, gene expression data, macromolecular structures, and proteomics data must be deposited in a publicly accessible database, and accession codes and associated hyperlinks must be provided in the “Data Availability” section.

Please include a “Data availability” subsection in the Online Methods. This section should inform readers about the availability of the data used to support the conclusions of your study, including accession codes to public repositories, references to source data that may be published alongside the paper, unique identifiers such as URLs to data repository entries, or data set DOIs, and any other statement about data availability. At a minimum, you should include the following statement: “The data that support the findings of this study are available from the corresponding author upon request”, describing which data is available upon request and mentioning any restrictions on availability. If DOIs are provided, please include these in the Reference list (authors, title, publisher (repository name), identifier, year). For more guidance on how to write this section please see: <http://www.nature.com/authors/policies/data/data-availability-statements-data-citations.pdf>

CODE AVAILABILITY

Please include a “Code Availability” subsection in the Online Methods which details how your custom code is made available. Only in rare cases (where code is not central to the main conclusions of the paper) is the statement “available upon request” allowed (and reasons should be specified).

For more information on our code sharing policy and requirements, please see: <https://www.nature.com/nature-research/editorial-policies/reporting-standards#availability-of-computer-code>

MATERIALS AVAILABILITY

ORCID

Best regards,
Nina

Nina Vogt, PhD
Senior Editor
Nature Methods

Reviewers' Comments:

[Redacted]

Decision Letter, first revision:

Dear Mickael,

Thank you for your letter detailing how you would respond to the reviewer concerns regarding your Article, "Functional Ultrasound Localization Microscopy reveals brain-wide neurovascular activity on a

microscopic scale". We have decided to invite you to revise your manuscript as you have outlined, before we reach a final decision on publication.

In the revised manuscript, please do add the analysis of CBF in different brain regions as well as the additional example of imaging in deep brain regions as shown in your response to my email. Please discuss the other issues brought up by the reviewer in the revision.

- * include a point-by-point response to the reviewers and to any editorial suggestions

- * please underline/highlight any additions to the text or areas with other significant changes to facilitate review of the revised manuscript

- * address the points listed described below to conform to our open science requirements

- * ensure it complies with our general format requirements as set out in our guide to authors at www.nature.com/naturemethods

- * resubmit all the necessary files electronically by using the link below to access your home page

[Redacted] This URL links to your confidential home page and associated information about manuscripts you may have submitted, or that you are reviewing for us. If you wish to forward this email to co-authors, please delete the link to your homepage.

We hope to receive your revised paper within 4 weeks. If you cannot send it within this time, please let us know. In this event, we will still be happy to reconsider your paper at a later date so long as nothing similar has been accepted for publication at Nature Methods or published elsewhere.

OPEN SCIENCE REQUIREMENTS

REPORTING SUMMARY AND EDITORIAL POLICY CHECKLISTS

Please note that these forms are dynamic 'smart pdfs' and must therefore be downloaded and completed in Adobe Reader. We will then flatten them for ease of use by the reviewers. If you would

like to reference the guidance text as you complete the template, please access these flattened versions at <http://www.nature.com/authors/policies/availability.html>.

DATA AVAILABILITY

Please include a “Data availability” subsection in the Online Methods. This section should inform readers about the availability of the data used to support the conclusions of your study, including accession codes to public repositories, references to source data that may be published alongside the paper, unique identifiers such as URLs to data repository entries, or data set DOIs, and any other statement about data availability. At a minimum, you should include the following statement: “The data that support the findings of this study are available from the corresponding author upon request”, describing which data is available upon request and mentioning any restrictions on availability. If DOIs are provided, please include these in the Reference list (authors, title, publisher (repository name), identifier, year). For more guidance on how to write this section please see: <http://www.nature.com/authors/policies/data/data-availability-statements-data-citations.pdf>

MATERIALS AVAILABILITY

ORCID

Best regards,

Nina

Nina Vogt, PhD

Senior Editor

Nature Methods

Reviewers' Comments:

Reviewer #1:

Remarks to the Author:

The authors have addressed all my concerns. I think it is an exceptionally nice manuscript, and I congratulate the authors on their work. I fully support its publication in Nature Methods.

Reviewer #2:

Remarks to the Author:

This paper, transferred from [Redacted], deals with a modification and technological advancement of ultrasound localization microscopy (ULM) using microbubbles (MB) to visualize blood flow in vessels. The authors were able to increase the spatial and temporal resolution of the method enabling the dynamic assessment of cerebrovascular flow in superficial and deep brain vessels. Therefore, changes in MB flow induced by cortical activation (whisker deflection or visual stimuli) can be simultaneously detected in pial arterioles at the brain's surface and in intraparenchymal arterioles deep in the cortex and in subcortical brain regions, like the thalamus and culliculi.

This method offers several advantages that may lead to advance the understanding of neurovascular coupling (NVC), including: dynamic assessment of flow in different vascular compartments, ability to monitor superficial and deep brain regions, sufficient resolution to monitor flow at the arteriolar to capillaries transition, and potential for non-invasive assessment not requiring a craniotomy. These characteristics are well suited to investigate the microvascular dynamics in the entire microvascular network as advocated in a recent critical review of the state-of-the-art in NVC.

However, there are also limitations in the method, some of which are mentioned in the text, but need to be more explicitly stated or addressed with experiments to provide proof-of-principle evidence of the full potential of the method to investigate NVC:

1. The data presented are in rat, a species that is no longer the first choice in neuroscience studies. Using this technique in mice would open the way to using genetically modified models and other molecular tools that are essential for mechanistic investigations of NVC.
2. The invasiveness of the large craniotomy is also a drawback. Transcranial imaging, which could be perfected (as indicated in the paper), would be a major advance, since competing imaging approaches currently used do not require craniotomy or can be applied to thin skull preparations.
3. The application to awake animals needs to be explored. Most recent studies on NVC have examined awake mice, since NVC is uniquely sensitive to anesthetics, which may lead to spurious results. The need for large volume injections and repetitive stimulation may also be limitations in awake behaving animals.
4. As noticed by one to the previous reviewers, the method involves injection of a large volume of fluid the impact of which on the physiological state of the animals is unclear. The assessment of heart rate and breathing is not sufficient for this purpose since cerebral blood flow is highly sensitive to changes in

blood pressure, blood gasses, hematocrit, blood volume, brain temperature, etc. Lacking a careful assessment of the impact of the fluid injection on these critical variables would preclude a correct interpretation of the changes in MB flow in hypothesis-testing situations.

5. The need to use repetitive stimulation to enhance SNR in slow flowing microvessels is also a limitation. More and more NVC is being studied during natural behaviors engaging the brain as a whole, which will be well suited to the present method which has the potential to image several brain regions at the same time.

6. To this end, more definitive evidence of the ability of the method to provide reliable vascular signals from slow flowing microvessels of deep brain regions would be desirable.

7. The heterogeneity of cerebral microvascular cells highlighted by single cell RNAseq studies requires microvascular assessment with cell-type specificity, which has an impact on microvascular function. The ability of fULM to monitor the full vascular network in combination with approaches to provide cell-type identification would provide a major advance to the field.

Reviewer #3:

Remarks to the Author:

I am satisfied with the authors' response to the questions raised. Only a very minor mistake to correct: in the manuscript "Positron Electron Tomography" were mentioned twice (page 2 and 38) but should it be "Position Emission Tomography"?

Author Rebuttal, first revision:

Reviewers' Comments:

Reviewer #1:

Remarks to the Author:

The authors have addressed all my concerns. I think it is an exceptionally nice manuscript, and I congratulate the authors on their work. I fully support its publication in Nature Methods.

We deeply thank reviewer 1 for his positive comments and his help to improve the final manuscript quality.

Reviewer #2:

Remarks to the Author:

This paper, transferred from [Redacted], deals with a modification and technological advancement of ultrasound localization microscopy (ULM) using microbubbles (MB) to visualize blood flow in vessels. The authors were able to increase the spatial and temporal resolution of the method enabling the dynamic assessment of cerebrovascular flow in superficial and deep brain vessels. Therefore, changes in MB flow induced by cortical activation (whisker deflection or visual stimuli) can be simultaneously detected in pial arterioles at the brain's surface and in intraparenchymal arterioles deep in the cortex and in subcortical brain regions, like the thalamus and culliculi.

This method offers several advantages that may lead to advance the understanding of neurovascular coupling (NVC), including: dynamic assessment of flow in different vascular compartments, ability to monitor superficial and deep brain regions, sufficient resolution to monitor flow at the arteriolar to capillaries transition, and potential for non-invasive assessment not requiring a craniotomy. These characteristics are well suited to investigate the microvascular dynamics in the entire microvascular network as advocated in a recent critical review of the state-of-the-art in NVC.

We deeply thank the reviewer 2 for these positive comments

However, there are also limitations in the method, some of which are mentioned in the text, but need to be more explicitly stated or addressed with experiments to provide proof-of-principle evidence of the full potential of the method to investigate NVC:

- 1. The data presented are in rat, a species that is no longer the first choice in neuroscience studies. Using this technique in mice would open the way to using genetically modified models and other molecular tools that are essential for mechanistic investigations of NVC.*

We thank the referee for this interesting comment. In fact, this technique could be applied straightforwardly in mice.

[Redacted]

2. The invasiveness of the large craniotomy is also a drawback. Transcranial imaging, which could be perfected (as indicated in the paper), would be a major advance, since competing imaging approaches currently used do not require craniotomy or can be applied to thin skull preparations.

Here, we are convinced that transcranial propagation should not be an issue. Indeed, we decided to perform a craniotomy in most of our experiments as this work corresponds to the proof of concept of a new methodology and we wanted to study the imaging method in optimal conditions. However, we did not sufficiently insist on the fact that the technique can also be applied in thinned skulls configurations or also in transcranial configurations. We agree that the experiments on transcranial fULM imaging in rats could be improved, but we have many other ongoing works (some published and some unpublished) with transcranial ultrasound localization microscopy showing that transcranial ULM imaging is feasible with convincing image quality. We now discuss carefully this point in the discussion part of the revised manuscript.

In order to support our opinion, we provide some further examples of images of transcranial ULM in rats and mice:

- In rats, a recent article was published by Chavignon et al (IEE TMI 2021), an independent group from a former member of our lab, in which a Raw Column Arrays (RCA) probe was used for transcranial ULM imaging in rats. The image below comes from this article (Chavignon et al, IEEE Transactions on Medical Imaging, 2021) and clearly shows that transcranial localization microscopy is feasible in rats.

[Redacted]

- Transcranial fULM will also strongly benefit from the addition of aberration corrections techniques and the further improvement of localization algorithms.

In conclusion, in the near future, these fULM experiments will be performed transcranially both in rats and mice.

3. *The application to awake animals needs to be explored. Most recent studies on NVC have examined awake mice, since NVC is uniquely sensitive to anesthetics, which may lead to spurious results. The need for large volume injections and repetitive stimulation may also be limitations in awake behaving animals.*

We are very confident that awake Functional ULM could be applied in further works on a head fixed experiments. Indeed, recent studies, using exactly the same probe and electronics, as the one used in these experiments, have shown that the sensitivity of functional ultrasound imaging of the brain activity (even without contrast agents) is sufficient to be used in transcranial + head fixed + awake configurations (See for example Bertolo et al, Whole-Brain 3D Activation and Functional Connectivity Mapping in Mice using Transcranial Functional Ultrasound Imaging, Journal of Visualized Experiments 2021 JOVE). Other independent groups are also performing head fixed awake functional ultrasound imaging with comparable technologies (see for example: Mace E. et al Whole-brain functional ultrasound imaging reveals brain modules for visuomotor integration, Neuron 100 (5), 1241-1251. e7. See also: Brunner C. et al, A platform for brain-wide volumetric functional ultrasound imaging and analysis of circuit dynamics in Awake Mice, Neuron 108 (5), 861-875. e7).

Note also that these publications performed in functional ultrasound imaging on awake mice imaging in a head fixed setup are performed with the same type of acquisition parameters, but without microbubbles. The application of transcranial functional ULM in mice would only require the simultaneous addition of microbubble injection.

[Redacted]

It would therefore be quickly adapted to future head fixed imaging experiments in awake mice.

In conclusion, such head fixed setups could be used to perform fULM in awake animals. This goes beyond the scope of the proof-of-concept paper, but it will be carried out in further works.

We agree this is an important point needs to be to discussed in our discussion. We have now added this point in the discussion of the revised manuscript.

4. *As noticed by one to the previous reviewers, the method involves injection of a large volume of fluid the impact of which on the physiological state of the animals is unclear. The assessment of heart*

rate and breathing is not sufficient for this purpose since cerebral blood flow is highly sensitive to changes in blood pressure, blood gasses, hematocrit, blood volume, brain temperature, etc. Lacking a careful assessment of the impact of the fluid injection on these critical variables would preclude a correct interpretation of the changes in MB flow in hypothesis-testing situations.

We agree this is an important point and we are now discussing this particular point both in the discussion and in the material and methods sections. As detailed below, we have strong arguments to suggest that the volume injected was moderate and had a limited impact on the physiological state of the animal and the measures performed.

1. First of all, international recommendations for the dose volume guidelines exists. The 3Rs Translational and Predictive Sciences Leadership Group - Contract Research Organization Working Group of IQ (international consortium for innovation and quality in pharmaceutical developments) has provided such international recommendations for the dose volume guidelines (document attached) based on an extensive literature. This document includes dose volume guidelines that have been researched and published as well as standards that have gained acceptance through empirical use across multiple members of the IQ 3Rs leadership group (LG) and partner CROs. The recommendation of this international committee for the maximal volume of intravenous injection in rats is 20 ml/kg with a slow injection (between 3 and 10 minute long). Our rats' weight was 300 g, meaning a maximum injection volume should not exceed 6.0 ml. In our experiments, we used a continuous slow injection at 3.5 ml/h during 20 min, corresponding to 1.1 ml (~1/6 of the maximum dose) at a rate 3 to 8 times slower than the slow injection described in these international recommendations (20 ml/kg in 5 to 10 minutes). Therefore, we humbly think that the total fluid injection is not a critical as it may seem.

2. Secondly, in addition to heart rate and breathing, we also have precise access in our data to possible changes of CBF during the experiment, by measuring the flow of microbubbles. Our results clearly show that the MB/s baseline does not vary significantly during the 20 minutes continuous injections. This was already shown in our results (in the supplementary figure 1). However, it is true that we did not sufficiently insist on the importance of this figure and its interpretation for this particular argument.

In the revised manuscript, the mention to this point and the supplementary fig. 1. were improved, by adding the baseline temporal signal of MB/s in different regions of the brain, rather than just the global

signal (see proposed suppl. Figure 1 below). We also provide further comments in the discussion section, reinforcing this point.

Furthermore, regarding your comment on the hematocrit, our injection volume of 1.1 ml for a 21 ml total blood volume (70 ml total blood volume for 300 g rat) corresponds only to a 5% change in the total blood volume, due to the injection, suggesting a very limited change in the hematocrit.

Supplementary figure1: Evolution of the Microbubbles flow injection profile. Continuous perfusion of MB provides a stable delivery over time: MBs detection count per ultrafast image (representative of the cerebral blood flow) for the whole brain, cortical and thalamic regions in a representative animal over the whole acquisition (23 minutes).

3. Finally, the fULM method is just at its early stages and there are many rooms of improvement to decrease the injection volume in the next years :

- on the processing side: to date, the number of detected bubbles per ultrafast image is typically $N=80$. Nevertheless, we keep only roughly 38% ($N\sim 30$) of these detected events during the tracking process. There is definitely some room for improvement in this high rejection rate. Increasing the number of detected microbubbles would allow us to decrease the injected volume.
- on the contrast agent side: the gas concentration of microbubbles used in these experiments was $5 \mu\text{l/ml}$. While keeping such very low gas concentration, the number of detectable microbubbles could be strongly increased by making them smaller. A decrease in diameter by a factor 2 would allow us to increase the number of microbubbles by a factor 8 with the same total gas content and injection volume. So, the size of microbubbles could be easily decreased from some micrometers down to 0.5-1

micrometer in order to strongly increase the number of MB per ml (one to two orders of magnitude) without increasing the gas content.

We propose to add some comments in the discussion and in the materials and methods of the manuscript and cite this document and other publications:

Recommended Dose Volumes for Common Laboratory Animals, International Consortium for innovation and Quality in pharmaceutical Developments, IQ 3R's Leadership Group- Contract Research Organization Working Group.

Diehl, K.H., Hull, R., Morton, D. Pfister, R., Rahemampianina, Y., Smith, D., Vidal, J-M., and Vorstenbosch, C. (2001). A Good Practice Guide to the Administration of Substances and Removal of Blood Including Routes and Volumes. *Journal of Applied Toxicology*. 21,15-23.

Morton D.B., Jennings, M., Buckwell, A., Ewbank, R., Godfrey, C., Holgate, B., Inglis, I., James, R., Page, C., Sharman, I., Verschoyle, R., Westall, L., and Wilson, A.B. (2001). Refining Procedures for the Administration of Substances. Report of the BVAAWF/FRAME/RSPCA/UFAW Joint Working Group on Refinement. *Laboratory Animals*. 35, 1-41.

5. *The need to use repetitive stimulation to enhance SNR in slow flowing microvessels is also a limitation. More and more NVC is being studied during natural behaviors engaging the brain as a whole, which will be well suited to the present method which has the potential to image several brain regions at the same time.*

As the reviewer knows, no brain imaging modality is able to image the functional blood flow variations at microscopic scale over the whole brain. Asking to perform this huge challenge (what our study is bringing forward) additionally for single trials (without stimulus repetition) and in natural behavior is asking to demonstrate a kind of perfect brain imaging modality. Every imaging modality has its own limitations and we always have to use different techniques to answer various scientific questions. One single modality cannot be suited to solve all the challenges of neuroimaging.

Of course, we agree that the need for repetitive stimuli is a limitation, however this limitation stands only for the smallest tiny vessels. For example, in typical 30 micrometer diameter arterioles, the number of microbubbles per second is sufficient to provide dynamic fULM even without repetition, i.e. in single trials. In smaller pre- capillary arterioles, the number of required repetitions is typically 5 and finally in the small higher order branches capillaries, the number of repetitions should be around 10 as Suppl. figure 9 shows, that the improvement from 10 repetitions to 20 repetitions remains marginal.

Yet, in functional neuroimaging, N=10 repetitions is not considered excessive. In fMRI and electrophysiology studies in behaving animals, tens of repetitions are very often required to provide significant results. This is even the reason why the sensitivity of functional Ultrasound has been shown to be so interesting as it enables to provide single trial experiments compared to fMRI and implanted electrodes recordings (see recent work: Dizeux A. et al Nature Comm 2019 in behaving primates) for cognitive studies in non-human primates and paves the way to Brain Machine Interfaces based on Brain ultrasound imaging (See Norman S. et al, 2021 Neuron).

Furthermore, although the stimuli repetition is necessary for tiny vessels, Fig. 4G shows that the signal processing based on the global SVD data decomposition enables to estimate the temporal profile of each single trial without requiring to perform the data averaging over repetitions.

Here, we managed to provide a functional response at the microscopic scale over the whole deep brain in a limited number of repetitions, similarly to many other functional imaging modalities. Although we fully respect the challenging and interesting comments of the reviewer, we must confess that we cannot solve all problems of functional brain imaging in a single paper.

6. *To this end, more definitive evidence of the ability of the method to provide reliable vascular signals from slow flowing microvessels of deep brain regions would be desirable.*

We agree that this would improve the manuscript.

We already provided images of vascular responses in the deep thalamic and colliculus regions. Results in fig. 3C and former fig 12C-D (corresponding to thalamic deep regions) clearly showed the ability of the

technique to quantitatively demonstrate reliable vascular measurements in small vessels. These vessels were about 30 μm diameter and it was possible to see the blood flow profile within the vessels.

In order to show the same in deep seated and very small microvessels, we also added a new supplementary figure (Suppl. Fig. 6) showing the vascular response in tiny vessels of the colliculus. For example, we can clearly see the functional vascular response in vessel θ that has a 15 μm diameter.

Supplementary figure 6: Temporal profile of the MB flow variation in pixels from deep seated microvessels of the colliculus during visual stimulation. (A) Location of pixel λ (in a 56 μm diameter vessel) and θ (in a 15 μm diameter vessel) in the superior colliculus. (B-C) Zoomed regions of interest from image (A). (D-E) Temporal profile of MB Count in pixels λ and θ before and after motion correction during the successive visual stimuli. (F) Temporal profile of MB Count in pixels λ and θ after pattern summation.

7. *The heterogeneity of cerebral microvascular cells highlighted by single cell RNAseq studies requires microvascular assessment with cell-type specificity, which has an impact on microvascular function. The ability of fULM to monitor the full vascular network in combination with approaches to provide cell-type identification would provide a major advance to the field.*

We thank the reviewer for this positive comment. It is true that the combination of fULM with single cell RNAseq studies is going to be an extremely powerful combination for Neuroscience community. We will mention the interest of this combination in the discussion.

Reviewer #3:

Remarks to the Author:

I am satisfied with the authors' response to the questions raised. Only a very minor mistake to correct: in the manuscript "Positron Electron Tomography" were mentioned twice (page 2 and 38) but should it be "Position Emission Tomography"?

We corrected this typo.

We deeply thank the reviewer for his positive comments and his help to improve the final quality of the manuscript.

Decision Letter, second revision:

Dear Mickael,

Thank you for submitting your revised manuscript "Functional Ultrasound Localization Microscopy reveals brain-wide neurovascular activity on a microscopic scale" (NMEMH-A46993B). It has now been seen by one of the original referees and their comments are below. The reviewer finds that the paper has improved in revision, and therefore we'll be happy in principle to publish it in Nature Methods, pending minor revisions to satisfy the referees' final requests and to comply with our editorial and formatting guidelines.

TRANSPARENT PEER REVIEW

Nature Methods offers a transparent peer review option for new original research manuscripts submitted from 17th February 2021. We encourage increased transparency in peer review by publishing the reviewer comments, author rebuttal letters and editorial decision letters if the authors agree. Such peer review material is made available as a supplementary peer review file. Please state in the cover letter 'I wish to participate in transparent peer review' if you want to opt in, or 'I do not wish to participate in transparent peer review' if you don't. Failure to state your preference will result in delays in accepting your manuscript for publication.

Thank you again for your interest in Nature Methods Please do not hesitate to contact me if you have any questions.

Best regards,

Nina

Nina Vogt, PhD

Senior Editor

Nature Methods

ORCID

Reviewer #2 (Remarks to the Author):

The authors have addresses my comments.

Final Decision Letter:

Dear Mickael,

I am pleased to inform you that your Article, "Functional Ultrasound Localization Microscopy reveals brain-wide neurovascular activity on a microscopic scale", has now been accepted for publication in Nature Methods. Your paper is tentatively scheduled for publication in our August print issue, and will be published online prior to that. The received and accepted dates will be September 1st, 2021 and June 14th, 2022. This note is intended to let you know what to expect from us over the next month or so, and to let you know where to address any further questions.

Your paper will now be copyedited to ensure that it conforms to Nature Methods style. Once proofs are generated, they will be sent to you electronically and you will be asked to send a corrected version within 24 hours. It is extremely important that you let us know now whether you will be difficult to contact over the next month. If this is the case, we ask that you send us the contact information (email,

phone and fax) of someone who will be able to check the proofs and deal with any last-minute problems.

If, when you receive your proof, you cannot meet the deadline, please inform us at rjsproduction@springernature.com immediately.

Once your manuscript is typeset and you have completed the appropriate grant of rights, you will receive a link to your electronic proof via email with a request to make any corrections within 48 hours. If, when you receive your proof, you cannot meet this deadline, please inform us at rjsproduction@springernature.com immediately.

Once your paper has been scheduled for online publication, the Nature press office will be in touch to confirm the details.

Content is published online weekly on Mondays and Thursdays, and the embargo is set at 16:00 London time (GMT)/11:00 am US Eastern time (EST) on the day of publication. If you need to know the exact publication date or when the news embargo will be lifted, please contact our press office after you have submitted your proof corrections. Now is the time to inform your Public Relations or Press Office about your paper, as they might be interested in promoting its publication. This will allow them time to prepare an accurate and satisfactory press release. Include your manuscript tracking number NMETH-A46993C and the name of the journal, which they will need when they contact our office.

About one week before your paper is published online, we shall be distributing a press release to news organizations worldwide, which may include details of your work. We are happy for your institution or funding agency to prepare its own press release, but it must mention the embargo date and Nature Methods. Our Press Office will contact you closer to the time of publication, but if you or your Press Office have any inquiries in the meantime, please contact press@nature.com.

Please note that Nature Methods is a Transformative Journal (TJ). Authors may publish their research with us through the traditional subscription access route or make their paper immediately open access through payment of an article-processing charge (APC). Authors will not be required to make a final decision about access to their article until it has been accepted. Find out more about Transformative Journals

Authors may need to take specific actions to achieve compliance with funder and institutional open access mandates. If your research is supported by a funder that requires immediate open access (e.g. according to Plan S principles) then you should select the gold OA route, and we will direct you to the compliant route where possible. For authors selecting the subscription publication route, the journal's standard licensing terms will need to be accepted, including self-archiving policies. Those licensing terms will supersede any other terms that the author or any third party may assert apply to any version of the manuscript.

To assist our authors in disseminating their research to the broader community, our SharedIt initiative provides you with a unique shareable link that will allow anyone (with or without a subscription) to read the published article. Recipients of the link with a subscription will also be able to download and print the PDF. As soon as your article is published, you will receive an automated email with your shareable link.

Please note that you and your coauthors may order reprints and single copies of the issue containing your article through Nature Research Group's reprint website, which is located at <http://www.nature.com/reprints/author-reprints.html>. If there are any questions about reprints please send an email to author-reprints@nature.com and someone will assist you.

Best regards,

Nina

Nina Vogt, PhD

Senior Editor

Nature Methods

** Visit the Springer Nature Editorial and Publishing website at www.springernature.com/editorial-and-publishing-jobs for more information about our career opportunities. If you have any questions please click here.**

This email has been sent through the Springer Nature Tracking System NY-610A-NPG&MTS

Confidentiality Statement:

This e-mail is confidential and subject to copyright. Any unauthorised use or disclosure of its contents is prohibited. If you have received this email in error please notify our Manuscript Tracking System Helpdesk team at <http://platformsupport.nature.com>.

Details of the confidentiality and pre-publicity policy may be found here
<http://www.nature.com/authors/policies/confidentiality.html>

Privacy Policy | Update Profile

DISCLAIMER: This e-mail is confidential and should not be used by anyone who is not the original intended recipient. If you have received this e-mail in error please inform the sender and delete it from your mailbox or any other storage mechanism. Springer Nature America, Inc. does not accept liability for any statements made which are clearly the sender's own and not expressly made on behalf of Springer Nature America, Inc. or one of their agents.

Please note that neither Springer Nature America, Inc. or any of its agents accept any responsibility for viruses that may be contained in this e-mail or its attachments and it is your responsibility to scan the e-mail and attachments (if any).

DISCLAIMER: This e-mail is confidential and should not be used by anyone who is not the original intended recipient. If you have received this e-mail in error please inform the sender and delete it from your mailbox or any other storage mechanism. Springer Nature America, Inc. does not accept liability for any statements made which are clearly the sender's own and not expressly made on behalf of Springer Nature America, Inc. or one of their agents.

Please note that neither Springer Nature America, Inc. or any of its agents accept any responsibility for viruses that may be contained in this e-mail or its attachments and it is your responsibility to scan the e-mail and attachments (if any).